# Stage-dependent differential influence of metabolic and structural networks on memory across Alzheimer's disease continuum

**Kok Pin Ng**[1,2,3†], **Xing Qian**[4†], **Kwun Kei Ng**[4], **Fang Ji**[4], **Pedro Rosa-Neto**[5,6], **Serge Gauthier**[7], **Nagaendran Kandiah**[3], **Juan Helen Zhou**[4,8,9]*, **Alzheimer's Disease Neuroimaging Initiative**

[1]Department of Neurology, National Neuroscience Institute, Singapore, Singapore; [2]Duke-NUS Medical School, Singapore, Singapore; [3]Lee Kong Chian School of Medicine, Nanyang Technological University Singapore, Singapore, Singapore; [4]Centre for Sleep and Cognition and Centre for Translational MR Research,Yong Loo Lin School of Medicine, National University of Singapore, Singapore, Singapore; [5]Translational Neuroimaging Laboratory, McGill University Research Centre for Studies in Aging, Alzheimer's Disease Research Unit, Douglas Research Institute, Le Centre intégré universitaire de santé et de services sociaux (CIUSSS) de l'Ouest-de-l'Île-de-Montréal, and Departments of Neurology, Neurosurgery, Psychiatry, Pharmacology and Therapeutics, McGill University, Montreal, Canada; [6]Montreal Neurological Institute, McGill University, Montreal, Canada; [7]Department of Neurology & Neurosurgery, McGill University, Montreal, Canada; [8]Department of Electrical and Computer Engineering, National University of Singapore, Singapore, Singapore; [9]Integrative Sciences and Engineering Programme (ISEP), National University of Singapore, Singapore, Singapore

*For correspondence:
helen.zhou@nus.edu.sg

†These authors contributed equally to this work

## Abstract

**Background:** Large-scale neuronal network breakdown underlies memory impairment in Alzheimer's disease (AD). However, the differential trajectories of the relationships between network organisation and memory across pathology and cognitive stages in AD remain elusive. We determined whether and how the influences of individual-level structural and metabolic covariance network integrity on memory varied with amyloid pathology across clinical stages without assuming a constant relationship.

**Methods:** Seven hundred and eight participants from the Alzheimer's Disease Neuroimaging Initiative were studied. Individual-level structural and metabolic covariance scores in higher-level cognitive and hippocampal networks were derived from magnetic resonance imaging and [18F] fluorodeoxyglucose positron emission tomography using seed-based partial least square analyses. The non-linear associations between network scores and memory across cognitive stages in each pathology group were examined using sparse varying coefficient modelling.

**Results:** We showed that the associations of memory with structural and metabolic networks in the hippocampal and default mode regions exhibited pathology-dependent differential trajectories across cognitive stages using sparse varying coefficient modelling. In amyloid pathology group, there was an early influence of hippocampal structural network deterioration on memory impairment in the preclinical stage, and a biphasic influence of the angular gyrus-seeded default mode

metabolic network on memory in both preclinical and dementia stages. In non-amyloid pathology groups, in contrast, the trajectory of the hippocampus-memory association was opposite and weaker overall, while no metabolism covariance networks were related to memory. Key findings were replicated in a larger cohort of 1280 participants.

**Conclusions:** Our findings highlight potential windows of early intervention targeting network breakdown at the preclinical AD stage.

**Funding:** Data collection and sharing for this project was funded by the Alzheimer's Disease Neuroimaging Initiative (ADNI) (National Institutes of Health Grant U01 AG024904) and DOD ADNI (Department of Defense award number W81XWH-12-2-0012). We also acknowledge the funding support from the Duke NUS/Khoo Bridge Funding Award (KBrFA/2019-0020) and NMRC Open Fund Large Collaborative Grant (OFLCG09May0035), NMRC New Investigator Grant (MOH-CNIG18may-0003) and Yong Loo Lin School of Medicine Research funding.

## Editor's evaluation

This paper presents important information about how potential network-based structural and metabolic imaging biomarkers are associated with memory performance during distinct disease stages, in line with previous hypothetical biomarker models. The study is conceptually sound and methodologically convincing and will be of interest to neuroscientists and medical professionals involved in the study of Alzheimer's disease and related neurodegenerative conditions.

## Introduction

Alzheimer's disease (AD) is a neurodegenerative disease that is characterised by neuropathological accumulation of amyloid-beta (Aβ) plaques (A), intraneuronal tau neurofibrillary tangles (T), and neurodegeneration (N) in the brain (*Braak and Braak, 1991*; *Serrano-Pozo et al., 2011*). While AD is traditionally a clinical-pathologic condition, the emerging development of biomarkers to profile AD pathophysiology has led to the proposal of AD as a biological construct based on the AT(N) system (*Jack et al., 2016*; *Jack et al., 2018*). The incorporation of the AT(N) classification into the clinical continuum will offer robust disease staging by combining both pathophysiological and cognitive phenotypes which span from cognitively intact to mild cognitive impairment (MCI) before progressing to the dementia stage (*Knopman et al., 2018*). Studies have suggested that Aβ is the first to become abnormal in AD, followed by downstream pathophysiological changes of tauopathy, neurodegeneration, and cognitive impairment (*Bateman et al., 2012*; *Jack et al., 2013a*; *Bertens et al., 2015*). While neurodegeneration is widely associated with worse cognitive impairment in neurocognitive disorders, it remains unknown whether the influence of neurodegeneration on cognitive function varies with AD biomarkers status and across the AD continuum.

Neurodegeneration represents neuronal injury in the forms of cerebral grey matter (GM) atrophy and hypometabolism. In AD, it is widely postulated that Aβ triggers tau-mediated toxicity leading to AD-type neurodegeneration in brain regions such as the hippocampus, the precuneus and posterior cingulate cortex (PCC), bilateral angular gyrus (ANG), and medial temporal lobes (*Chételat et al., 2008*; *Misra et al., 2009*; *Mosconi et al., 2009*; *Kljajevic et al., 2014*). Recently, amyloid and tau pathologies are also shown to have a synergistic effect on AD-type hypometabolism, involving the basal and mesial temporal, orbitofrontal, and anterior and posterior cingulate cortices (*Hanseeuw et al., 2017*; *Pascoal et al., 2017*). However, neurodegeneration may also occur prior to incident amyloid positivity (*Jack et al., 2013b*) and be influenced by the loss of microtubule stabilising function and toxic effects of tau pathology, independent of amyloid pathology (*Ballatore et al., 2007*).

Advancement in brain network analysis offers insights into the functional effects of AD pathophysiology on cognitive changes. Work from our group has demonstrated that AD pathophysiologies compromise brain structure and function systematically by capitalising on the intrinsic connectivities among brain regions (*Zhou et al., 2012*). Accumulating evidence suggests that AD pathological deposition around neurons which impairs synaptic communication, leads to specific large-scale brain intrinsic network disorganisation (*Seeley et al., 2009*; *Marchitelli et al., 2018*). Decreased functional connectivity in the default mode network (DMN) derived from resting state functional MRI is well-described in MCI and AD (*Greicius et al., 2004*; *Zhou et al., 2010*; *Chong et al., 2017*; *Chong et al.,*

2019; *Zhou et al., 2017*), while aberrant loss of functional connectivity in other higher-order cognitive networks such as the executive control network (ECN) and salience network (SN) are being increasingly reported (*Chong et al., 2017*; *Brier et al., 2012*; *He et al., 2014*).

Brain networks can also be constructed based on similarity in GM structure and metabolism between brain areas across individuals, known as the GM structural and metabolic covariance network, respectively (*Ripp et al., 2020*; *Zielinski et al., 2010*; *Montembeault et al., 2012*). Both structural and metabolic covariance networks show convergent patterns with the intrinsic connectivity network in healthy individuals and mirror GM atrophy patterns in distinct neurodegenerative disorders (*Seeley et al., 2009*; *Ripp et al., 2020*; *Lizarraga et al., 2021*). Using this approach, a recent study revealed differential patterns of structural covariance networks within different amyloid pathology groups classified by cerebrospinal fluid (CSF) Aβ$_{1-42}$ and P-tau$_{181}$ levels (*Li et al., 2019*). However, existing studies on the GM structural and metabolic covariance networks were largely reliant on group-level correlation maps of cortical morphology and metabolism, which cannot be used to infer individual differences in cognition. It is postulated that network analysis at the individual level will allow direct evaluation of each individual's structural and metabolic covariance networks, hence providing deeper understanding on the effects of brain networks on cognitive performances (*Kim et al., 2016*). For instance, a cube-based correlation approach to calculate the individual GM networks by computing intracortical similarities in GM morphology (*Tijms et al., 2012*) showed that single-subject GM graph properties were associated with individual differences of clinical progression in AD (*Tijms et al., 2018*; *Tijms et al., 2014*; *Vermunt et al., 2020*; *Tijms et al., 2013*). A network template perturbation approach was also introduced to construct an individual differential SCN using regional GM volume, though it required reference models derived from a group of normal control subjects (*Liu et al., 2021*). Nevertheless, the relationships between changes in individual-level network-based neurodegeneration across different amyloid pathology groups and cognitive stages, and their influence on memory impairment, remain unclear.

The influence of cerebral GM loss and [$^{18}$F] fluorodeoxyglucose (FDG) hypometabolism on cognitive function in AD has often been modelled as a linear relationship (*Habeck et al., 2012*; *Bejanin et al., 2017*). However, emerging evidence suggests that structural and metabolic abnormalities in AD may follow a sigmoidal curve trajectory with an initial period of acceleration and subsequent deceleration (*Jack et al., 2013a*; *Sabuncu et al., 2011*; *Schuff et al., 2012*). While the dynamic effects of AD biomarkers on worsening cognition can be better modelled by sigmoid-shaped curves rather than a constant across disease stages (*Caroli et al., 2010*), it remains largely unknown how brain structural and metabolic networks will influence cognition decline differentially in individuals stratified into different pathology groups and cognitive stages. Once these trajectories are defined across the AD continuum and subgroups, they can potentially highlight windows of opportunity for targeted intervention at the appropriate cognitive stages to improve disease outcomes.

To cover these gaps, we sought to determine the differential associations of brain metabolism and GM structural networks with memory function using a neurodegeneration covariance network approach, among cognitively normal (CN), MCI, and probable AD individuals stratified by their A and T biomarker status. We used the seed partial least squares (PLS) method (*Krishnan et al., 2011*) to evaluate the individual-level brain network integrity. We employed the sparse varying coefficient (SVC) model which does not assume a constant relationship between brain measures and cognitive performance over different cognitive stages (*Hong et al., 2015*; *Daye et al., 2012*; *Ji et al., 2019*). Besides capturing the possible non-linear brain-cognition relationship, SVC also allows the selection of significant predictors with the least absolute shrinkage and selection operator (LASSO) sparse penalty while eliminating the contribution of the less important predictors. We hypothesised that individual-level brain metabolic and structural network integrity would be non-linearly associated with memory performance across the AD continuum and such trajectories would vary depending on the presence of amyloid and tau protein deposition. Based on our previous findings (*Zhou et al., 2010*; *Chong et al., 2017*; *Zhang et al., 2020*), we further hypothesised that the posterior DMN and the medial temporal lobe regions would play an early and dominant role affecting the memory performance in individuals with amyloid pathology.

Our study provides first evidence that both hippocampal structural and ANG metabolic network integrity contributed to memory performance in the early cognitively normal stage in individuals with amyloid deposition. However, in the amyloid positive individuals with dementia, only the ANG

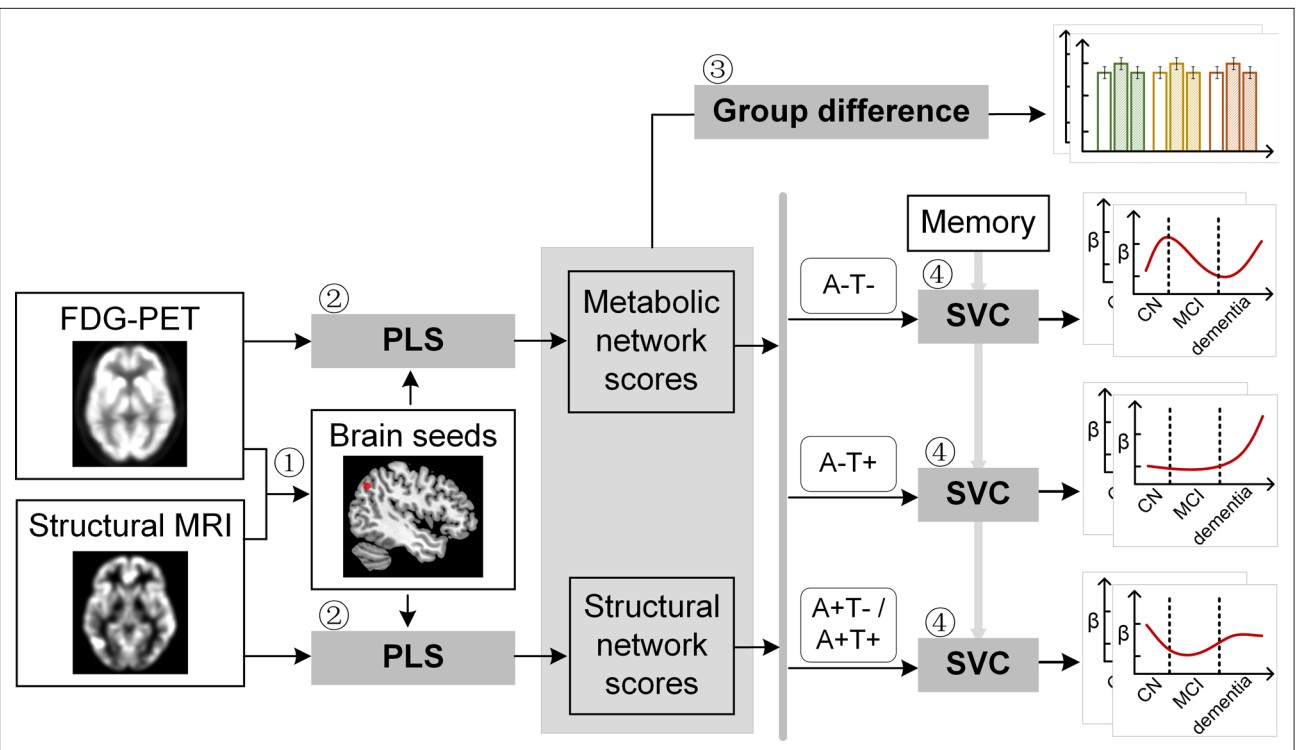

**Figure 1.** Study design schematic. Seven hundred and eight participants with either healthy cognition (CN), mild cognitive impairment (MCI) or dementia were studied. Twelve brain seeds covering the key regions of hippocampus, the default mode network, the executive control network, and salience network were defined based on hypometabolism (via FDG) and grey matter atrophy (via MRI) patterns in all patients with probable AD compared to CN (step 1). Using seed-based partial least square (PLS) analysis (step 2), the covariance patterns in metabolism and grey matter volume maps were identified and used to derive the individual-level brain metabolic network scores and structural network scores for each seed. The group difference was evaluated between different cognitive stages and pathology groups (step 3). We then investigated the differential stage-dependent associations between these key brain network scores with memory performance in each of the three pathology groups (A-T-, A-T+, and A+T-/A+T + ) separately using sparse varying coefficient (SVC) modelling (step 4). Abbreviations: A=Aβ; T=tau; '-' = negative; '+' = positive.

The online version of this article includes the following figure supplement(s) for figure 1:

**Figure supplement 1.** Flowchart of participant pool selection.

**Figure supplement 2.** Subject ordering for SVC modelling within each pathology group.

metabolic network dominated the memory-network association. Amyloid negative individuals did not have such patterns. These findings characterise the dynamic influence of brain structural and metabolic networks on memory function across the AD continuum and underscore the importance of early intervention targeting neuronal dysfunction in the preclinical AD stage to improve memory outcomes.

## Results

### Group differences in brain metabolic and structural covariance networks

We selected 812 participants (232 CN, 413 MCI, and 167 probable AD) from the Alzheimer's Disease Neuroimaging Initiative (ADNI) database with 3T T1-weighted MRI and [¹⁸F]FDG PET scans to define seed regions for brain network derivation (*Figure 1*, step 1 and *Figure 1—figure supplement 1*). As our study focused on memory and AD pathology, we chose to study the individual-level structural and metabolic covariance within higher-order cognitive networks such as DMN, SN, ECN as well as the hippocampus (HIP)-based memory network (*Veldsman et al., 2020*; *Vincent et al., 2008*). We defined a set of 12 seed regions to derive these covariance networks on the basis that they have been shown to reliably produce the relevant network across imaging modalities. Specifically, the DMN included bilateral ANG, PCC, and medial prefrontal cortex (mPFC); the SN included bilateral anterior insular (INS); the ECN included bilateral dorsolateral prefrontal cortex (DLPFC) and posterior parietal

cortex (PPC); the memory network included bilateral HIP. The seed coordinates were determined based on the group comparisons of the grey matter volume (GMV) probability and glucose metabolic spatial maps between CN and probable AD individuals (*Supplementary file 1* and *Supplementary file 4*, see details in Methods).

To derive brain structural and metabolic networks from individuals with and without amyloid pathology, we further identified 708 out of the existing 812 participants who underwent neuropsychological assessments, and lumbar puncture, in addition to [$^{18}$F]FDG PET and T1-weighted MRI scans to form the main dataset (*Table 1*). Using seed PLS (*Figure 1*, step 2, see details in Methods), we identified the structural and metabolic covariance network patterns associated with each seed at the group-level (*Figure 2A* and *Figure 3A*). We projected the original individual GMV and metabolic maps onto the covariance network maps to derive the individual brain structural or metabolic network scores, which reflected how strongly each brain network pattern was manifested in the individual's metabolic and structural brain networks.

First, we compared the brain metabolic and structural network scores between different pathology groups and cognitive stages (*Figure 1*, step 3). In participants with amyloid pathology (A+T-/A+T +), the probable AD group had lower metabolic and structural network scores than the CN and MCI groups in all the networks (*Figure 2B* right and 3B right). No such difference was observed in participants without amyloid pathology.

At the same cognitive stage, we observed slightly different patterns in structural and metabolic networks. Specifically, at the same cognitive stage, amyloid positive (A+T-/A+T + ) MCI individuals had lower metabolic and structural network scores than the MCI individuals without amyloid and tau pathology (A-T-) for all the networks (*Figures 2B and 3B*). The amyloid positive CN individuals had comparable structural network scores but lower metabolic network scores than the CN individuals without amyloid and tau pathology. In contrast, CN individuals with amyloid pathology (A+T-/A+T + ) showed lower structural integrity in the HIP-based memory network, the mPFC-based anterior DMN and the INS-based SN than the CN individuals with tau pathology only (A-T+). In addition, CN individuals with tau pathology (A-T+) had lower structural mPFC-based anterior DMN scores than the CN group without tau and amyloid pathology (A-T-).

## Divergent stage-dependent trajectories of the association between hippocampal structural network integrity and memory performance in the three pathology groups

Next, we sought to determine the differential non-linear trajectories of the association between brain network integrity and memory impairment in different pathology groups across the three cognitive stages using the SVC model (*Figure 1*, step 4; *Hong et al., 2015*). Note that we did not assume a constant relationship here; instead, the network-memory association could vary across cognitive stages. Instead of analysing each brain measure in separate models, the SVC analysis allows all variables to be entered as predictors in the same multivariate model, with the identification of the most important predictors and the elimination of the less important predictors (i.e. feature selection) implemented by minimising the penalised least squares function.

To characterise the possible stage-dependent trajectories using SVC modelling, we ordered the participants by their cognitive stages (i.e. CN → MCI → dementia; *Figure 1—figure supplement 2A*) in each of the three pathology groups (A-/T-, A-/T+and A+T-/A+T + ). Within each stage, the participants were then ordered by their global cognition or dementia severity (i.e. no impairment → severe impairment). Specifically, the participants within the CN group were ordered by decreasing MMSE scores, while the participants within the MCI and dementia groups were ordered by increasing CDR-sum of boxes (SOB) scores. Participants with the same MMSE or CDR-SOB scores were further ordered by increasing age (i.e. young → old). Ordered participants were distributed evenly into bins (i.e. 10 subjects/bin). In our SVC models, the dependent variable was the ADNI memory composite score. Predictors included all the 14 FDG/GMV regional network scores with gender, education years, *APOE* ε4, intracranial volume (ICV), and scanning site as nuisance variables. We performed the SVC modelling for each pathology group separately to find the key predictors and the trajectories of their associations with memory along the disease progression (see details in Methods).

The SVC models identified the HIP-based structural memory network score as a key predictor of memory impairment in all three pathology groups (*Figures 4A and 5A*). We found that the lower HIP

**Table 1.** Subject demographics for the main study cohort.

| | A-T- | | | A-T+ | | | A+T-/A+T+ | | |
|---|---|---|---|---|---|---|---|---|---|
| | CN | MCI | probable AD | CN | MCI | probable AD | CN | MCI | probable AD |
| N | 30 | 74 | 4 | 80 | 75 | 7 | 85 | 225 | 128 |
| Age, years | 65.12~85.16 | 56.08~88.51 | 69.56~90.50 | 56.53~84.47 | 55.15~88.83 | 60.79~80.76 | 60.19~90.08 | 55.38~91.57 | 55.96~90.46 |
| | 72.24±4.54 | 69.78±7.20[d] | 77.37±9.11[m] | 71.95±5.87 | 70.21±8.16 | 74.57±7.70 | 75.37±6.59 | 73.17±6.93 | 74.12±8.19 |
| Gender (M/F) | 12/18 | 39/35 | 4/0 | 45/35 | 38/37 | 6/1 | 37/48 | 127/98 | 72/56 |
| Handedness (R/L) | 29/1 | 60/14 | 4/0 | 69/11 | 67/8 | 7/0 | 79/6 | 203/22 | 118/10 |
| Education, years | 16.63±2.68 | 16.45±2.59 | 17.50±1.29 | 16.88±2.67 | 16.00±2.65 | 16.71±2.43 | 16.34±2.36 | 16.12±2.74 | 15.78±2.71 |
| APOE e4 (+/-) | 6/24 | 16/58 | 0/4 | 16/64 | 17/58 | 1/6 | 38/47[md] | 140/85[cd] | 93/35[cm] |
| Memory | 1.28±0.66[md] | 0.75±0.64[cd] | -0.12±0.68[cm] | 1.17±0.57[md] | 0.55±0.62[cd] | -0.40±0.67[cm] | 0.97±0.63[md] | 0.20±0.63[md] | -0.87±0.53[cm] |
| MMSE | 28.70±1.68[d] | 28.62±1.31[d] | 25.75±2.36[cm] | 29.15±0.99[md] | 28.29±1.64[cm] | 23.86±2.19[cm] | 29.02±1.17[md] | 27.81±1.86[cd] | 23.21±2.24[cm] |
| CDR-SOB | 0.02±0.09[md] | 1.22±0.60[cd] | 4.38±3.04[cm] | 0.05±0.15[md] | 1.20±0.76[cd] | 4.57±1.40[cm] | 0.05±0.17[md] | 1.53±0.90[cd] | 4.64±1.70[cm] |
| ICV | 1523.77±152.57 | 1520.05±127.78 | 1540.83±36.50 | 1554.01±128.10 | 1559.14±147.96 | 1566.75±221.76 | 1524.87±148.29 | 1558.61±148.25 | 1549.78±163.05 |

Note: Data on age are range and mean ± SD. Data on education, ICV, and memory are mean ± SD. Data on memory are in z-scores. Abbreviations: CN = cognitively normal; MCI = mild cognitive impairment; AD = Alzheimer's disease; A = β-amyloid; T = tau; '+' = positive; '-' = negative; y = years; M = male; F = female; R = right; L = left; MMSE = Mini-Mental State Exam; CDR-SOB = Clinical Dementia Rating Sum of Box; ICV = intracranial volume. Superscripts ([c], [m], [d]) represent significant group difference with CN, MCI and probable AD, respectively.

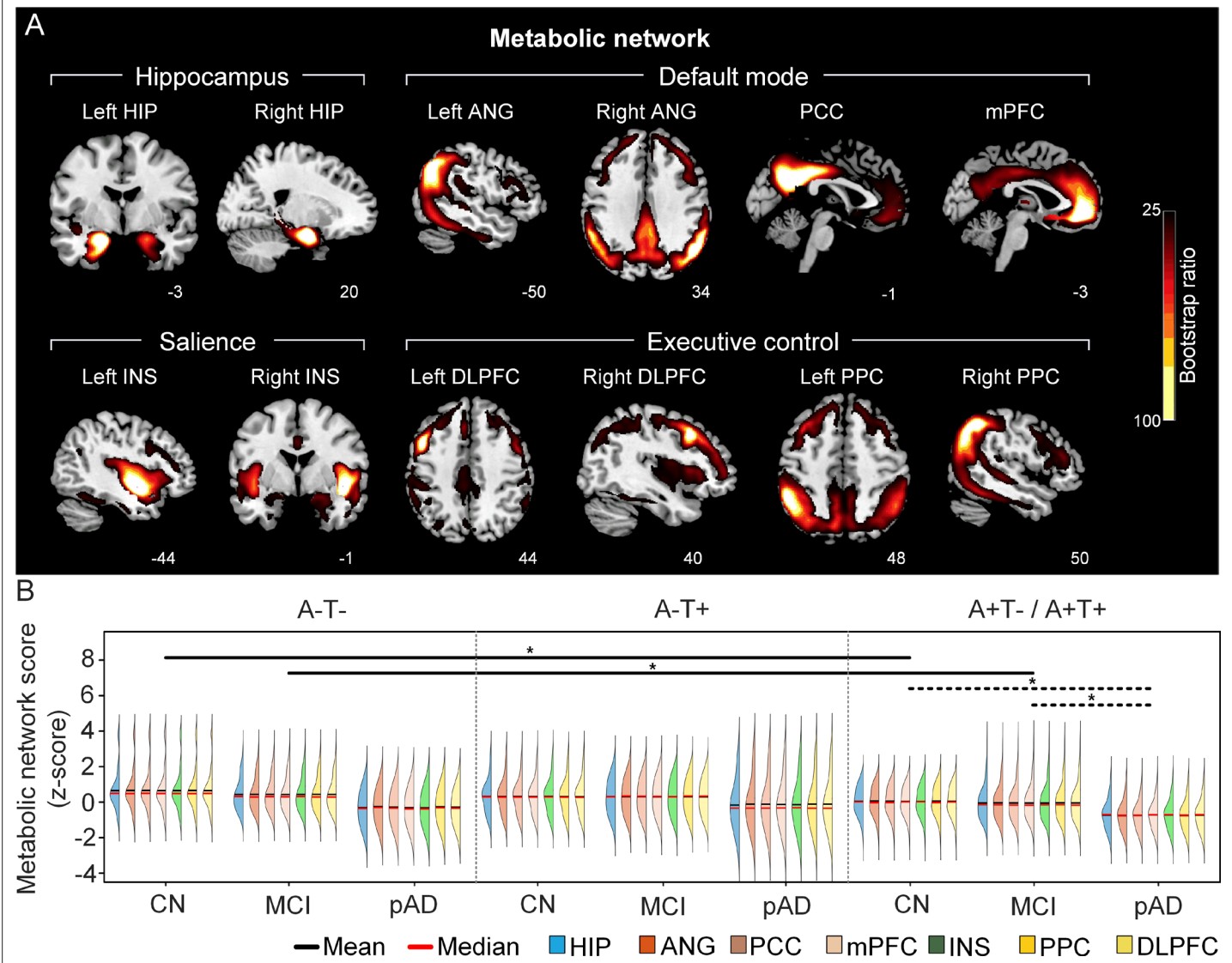

**Figure 2.** The integrity of brain metabolic networks in participants with and without amyloid pathology across cognitive stages. (A) Brain slices of metabolic covariance networks associated with each brain seed. Brain metabolic network resemabled canonical brain networks. The intensity of colorbar represents bootstrap ratios, derived from dividing the weight of the singular-vector by the bootstrapped standard error. (B) Individual-level brain metabolic network scores (z-score) were lower in individuals with worse cognition and amyloid pathology. Z-scores were calculated within all the subjects. Summary of individual-level metabolic network scores (mean and median) were presented in half-violin plots. '*' indicates significant group difference (p<0.05). Thick lines indicate group differences in brain scores of all the seven networks between different cognitive stages (grey dashed lines) or pathology groups (dark lines). Abbreviations: HIP = hippocampus; ANG = angular gyrus; PCC = posterior cingulate cortex; mPFC = media prefrontal cortex; INS = insular; DLPFC = dorsolateral prefrontal cortex; PPC = posterior parietal cortex; CN = cognitively normal; MCI = mild cognitive impairment; pAD = probable AD; A = β-amyloid; T=tau; '+'=positive; '-'=negative.

The online version of this article includes the following figure supplement(s) for figure 2:

**Figure supplement 1.** The integrity of brain metabolic networks in participants with and without amyloid pathology across cognitive stages (validation dataset 1).

structural network scores, the lower the ADNI-mem scores (indicated by positive beta coefficient). The strength of this association was higher (i.e. higher beta coefficient) in the amyloid pathology group than the other two A-groups.

More importantly, not only was the relationship between the HIP structural network and memory performance non-linearly dependent on cognitive stages as hypothesised, but such non-linear trajectories were also different across the three pathology groups (*Figures 4A and 5A*). Specifically, in

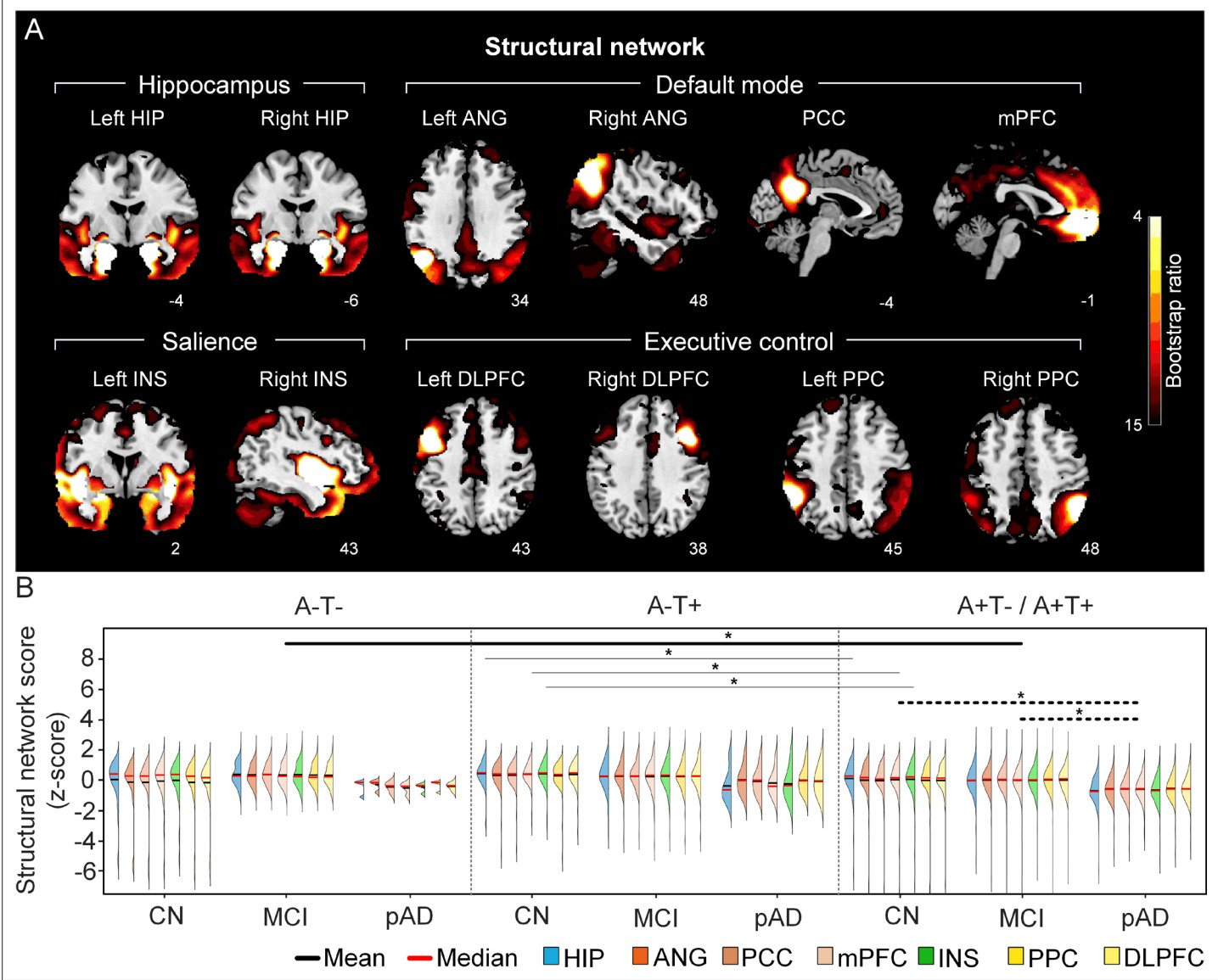

**Figure 3.** The integrity of brain structural networks in participants with and without amyloid pathology across cognitive stages. (A) Brain slices of structural covariance networks associated with each brain seed. The intensity of colorbar represents bootstrap ratios, derived from dividing the weight of the singular-vector by the bootstrapped standard error. (B) Individual-level brain structural network scores (z-score) were lower in individuals with worse cognition and amyloid pathology. Z-scores were calculated within all the subjects. Summary of individual-level structural network scores (mean and median) were presented in half-violin plots. '*' indicates significant group difference (p<0.05). Thick lines indicate group differences in brain scores of all the networks between different cognitive stages (grey dashed lines) or pathology groups (dark lines). Thin lines indicate group differences in brain scores of specific networks. Abbreviations: HIP = hippocampus; ANG = angular gyrus; PCC = posterior cingulate cortex; mPFC = media prefrontal cortex; INS = insular; DLPFC = dorsolateral prefrontal cortex; PPC = posterior parietal cortex; CN = cognitively normal; MCI = mild cognitive impairment; pAD = probable AD; A = β-amyloid; T=tau; '+' = positive; '-' = negative.

The online version of this article includes the following figure supplement(s) for figure 3:

**Figure supplement 1.** The integrity of brain structural networks in participants with and without amyloid pathology across cognitive stages (validation dataset 1).

the amyloid pathology group, the strength of this association was highest in early CN stage, and decreased from late CN to early MCI stage (*Figure 4A*, left). The strength of this association remained stable in MCI and then decreased in the dementia stage.

The two amyloid negative groups had the opposite pattern of the amyloid positive group (*Figure 5A*). Specifically, in the A-T- group, the strength of the association between the HIP structural

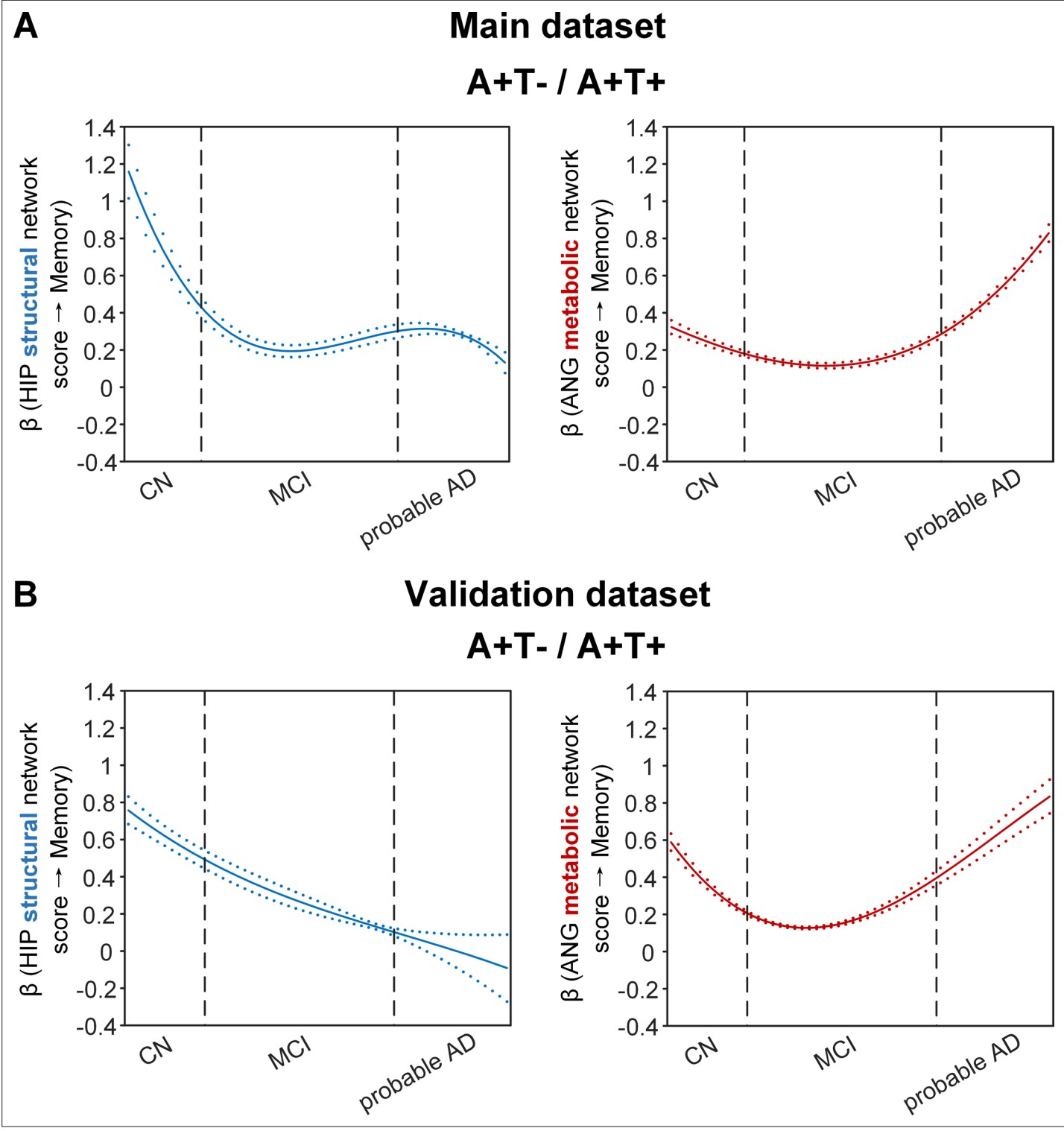

**Figure 4.** Brain metabolic and structural networks had differential stage-dependent associations with memory in amyloid positive individuals. Data from the main dataset (panel A) and validation dataset 1 (panel B) exhibited consistent stage-dependent memory-network association trajectory from cognitively normal to dementia stage in participants with amyloid pathology (i.e. A+T-/A+T + group). Both hippocampal-seeded structural network (left, in blue) and angular gyrus-seeded default mode metabolic network (right, in red) integrity contributed significantly to memory performance in early cognitively normal stage. Such impact decreased in MCI stage for both metabolic and structural networks. In contrast, only the metabolic network had a major influence on memory in late dementia stage. Solid curves represent the mean associations (beta coefficients) of brain network scores with memory as a function of advancing AD continuum estimated from 100 replicates. The dashed curves represent the point-wise 2* standard errors of the solid

*Figure 4 continued*

curves estimated from 100 replicates. The participants were ordered by their cognitive stages (i.e. CN → MCI → probable AD). Within each cognitive stage, the participants were then ordered by general cognition (MMSE for CN) or dementia severity (CDR for MCI and dementia) (i.e. no impairment → severe impairment). Participants with the same level of impairment/severity were further ordered by increasing age (i.e. young → old). Ordered participants were distributed evenly into bins (i.e. 10 subjects/bin). Abbreviations: CN = cognitively normal; MCI = mild cognitive impairment; HIP = hippocampus; ANG = angular gyrus.

The online version of this article includes the following figure supplement(s) for figure 4:

**Figure supplement 1.** Differential stage-dependent associations of metabolic and structural network scores with memory impairment in amyloid positive individuals (validation analyses).

network and memory performance was lowest in early CN stage and increased in the late CN stage. It then remained stable in the MCI stage before a further increase in the dementia stage. Similarly, in the A-T +group, the strength of such association was low in the CN stage and gradually increased in the MCI stage, reaching the highest in the late MCI and dementia stage.

Our findings suggest that the influence of the HIP-based structural network integrity on memory performance begins early in the preclinical AD stage and the strength of this influence gradually decreased as the cognitive stages progress. On the other hand, the influence of the HIP network integrity on memory is weaker in individuals without Aβ pathology and peaks in the dementia stage. The stronger hippocampus-memory association in the preclinical AD stage supports the current strategy of early intervention to attain better cognitive outcomes.

Furthermore, demographical and genetic variables such as gender, education years and *APOE* ε4 genotype showed differential stage- and pathology-dependent associations with memory performance (*Figure 5—figure supplement 2*). Females and fewer years of education were associated with memory impairment in A-/T- and A-/T+groups respectively. These associations were highest in the early CN stage and gradually decreased in late CN stage before increasing in the late MCI and probable AD stages. In contrast, females, fewer years of education and *APOE* ε4 carriers in the amyloid pathology group were associated with memory impairment with a differential trajectory (i.e. highest in the early CN stage and gradually decreased afterwards), although the strength of this association was relatively lower overall compared to those in the A-/T- and A-/T+groups.

## Stage-dependent association between angular gyrus metabolic network integrity and memory performance in amyloid pathology group

The SVC models identified the ANG-based metabolic network score (i.e. DMN) to be associated with memory impairment only in the amyloid pathology group (*Figure 4A*, right). We found that the lower the ANG metabolic network score, the lower the ADNI-mem score. This suggested that a breakdown in the ANG-based metabolic covariance network was related to worse memory performance in the amyloid pathology group only. A non-linear relationship was also observed between the ANG metabolic covariance network and memory performance across different cognitive stages. The strength of this relationship showed an early peak in early CN and gradually decreased in the late CN and MCI stages, before increasing in late MCI/dementia stage again.

Our findings are in line with the current literature which show that decreased glucose uptake in the ANG is associated with worse cognitive performance in the later stages of AD. In addition, we extend this field by demonstrating the early influence of the ANG-based metabolic covariance network (mirroring the DMN) for memory performance in the preclinical AD stage. This suggests that early metabolic dysfunction of the ANG and the extended DMN may predispose individuals with preclinical AD to be more vulnerable to memory impairment.

## Replication in the validation datasets

To test if the above findings from the main dataset can be replicated, we repeated the same analyses using a larger validation dataset (here after refer as validation dataset 1). We added an additional 468 individuals who underwent 1.5T T1-weighted MRI scans and [$^{18}$F]FDG PET. With the original main dataset of 812 participants, we had 1280 participants in total for brain seed definition (*Figure 1*, step 1). Out of 1280 participants, 859 participants had the same neuropsychological assessments,

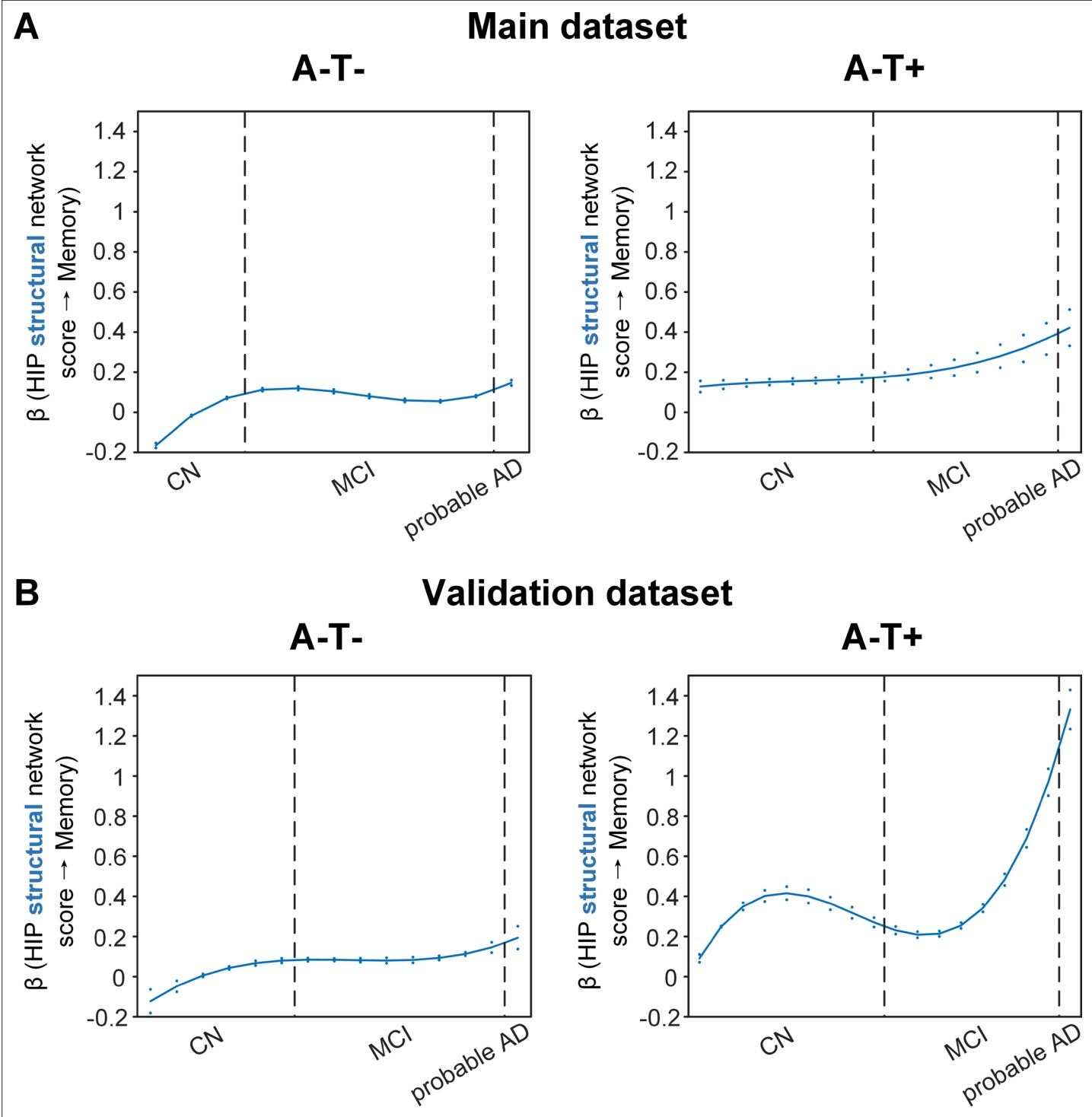

**Figure 5.** Stage-dependent association of brain hippocampal structural network with memory performance in A-T- and A-T +pathology groups. Data from the main dataset (panel A) and validation dataset 1 (panel B) exhibited consistent stage-dependent memory-network association trajectory from cognitively normal stage to dementia stage in participants with A-T- and A-T +pathology. The hippocampus-memory association was much weaker overall in non-amyloid/non-tau and tau only groups compared to amyloid positive group (*Figure 4*). The memory-network association was the lowest in early cognitively normal stage and gradually increased with clinical progression in both groups, while the tau only group had stronger associations in dementia stage. Solid curves represent the mean associations (beta coefficients) of brain network scores with memory as a function of advancing AD continuum estimated from 100 replicates. The dashed curves represent the point-wise 2* standard errors of the solid curves estimated from 100 replicates. The participants were ordered by their cognitive stages (i.e. CN → MCI → probable AD). Within each cognitive stage, the participants were then ordered by general cognition (MMSE for CN) or dementia severity (CDR for MCI and dementia) (i.e. no impairment → severe impairment).

*Figure 5 continued on next page*

*Figure 5 continued*

Participants with the same level of impairment/severity were further ordered by increasing age (i.e. young → old). Ordered participants were distributed evenly into bins (i.e. 10 subjects/bin). Abbreviations: CN = cognitively normal; MCI = mild cognitive impairment; HIP = hippocampus.

The online version of this article includes the following figure supplement(s) for figure 5:

**Figure supplement 1.** Differential stage-dependent associations of metabolic and structural network scores with memory impairment in A-T- and A-T +pathology groups (validation analyses).

**Figure supplement 2.** Differential stage-dependent associations of demographic variables with memory impairment in different pathology groups (main dataset).

**Figure supplement 3.** Differential stage-dependent associations of demographical variables with memory impairment in different pathology group (validation dataset 1).

**Figure supplement 4.** Variable selection frequency distribution for permuted datasets using sparse varying-coefficient (SVC) model.

lumbar puncture for the following analyses (*Figure 1*, steps 2 and 3, *Supplementary file 2*). The field strength (i.e. 1.5 T or 3 T) was further included as an additional nuisance variable for analyses on the validation dataset 1. We performed the same PLS-SVC analyses on validation dataset 1 and replicated most of our key findings (*Figures 4B and 5B*, *Figure 2—figure supplement 1*, *Figure 3—figure supplement 1* and *Figure 5—figure supplement 3*). Specifically, the HIP-based structural memory network and the ANG-based metabolic DMN scores were associated with memory impairment in the respective pathology groups with similar beta curves as the main dataset. Furthermore, these observations remained robust when the analyses were performed using the alternative ordering strategy of merging both MCI and dementia stages (*Figure 4—figure supplement 1C* & *Figure 5—figure supplement 1B*). Moreover, we also repeated the same analyses using only the 468 independent individuals (here after refer as validation dataset 2; *Figure 1—figure supplement 1*). Due to the small sample sizes of the A-T- and A-T +groups, we only performed the SVC modelling on the A+T-/A+T + group, which revealed consistent findings (i.e. predictors and stage-dependent trajectories) as the main dataset (*Supplementary file 1*, *Supplementary file 3* and *Supplementary file 4*; *Figure 4—figure supplement 1A*).

To test whether the results were sensitised to the relative imbalance of group sizes across diagnoses, we repeated the same analyses in the validation dataset 1 by performing PLS on the CN group only to generate the group-level brain salience maps in a healthy cohort. The individual brain glucose metabolic and grey matter probably spatial maps were then projected onto these CN-derived salience maps to generate the individual brain network scores. The subsequent SVC modelling replicated most of our key findings from the main dataset (*Figure 4—figure supplement 1B* and *Figure 5—figure supplement 1A*).

## High specificity of the SVC model

Last, we evaluated the specificity of the established SVC models using permutation tests. For each null SVC model using the permuted datasets, the frequency distributions of variable selection (i.e. the total times of selection as the key predictor of memory scores within the 100 permuted datasets) appeared random (*Figure 5—figure supplement 4*). As the selected variables in our main findings were not favoured over the other variables in the null models, this indicated the high specificity of the SVC models that were built on the original dataset. To further evaluate the specificity and robustness of the SVC models, we replaced LASSO with Ridge as the penalty in the SVC modelling. All results obtained on the main dataset with either LASSO or Ridge as the penalty in the SVC modelling were consistent (*Figure 4—figure supplement 1D* & *Figure 5—figure supplement 1C*).

## Discussion

This study revealed differential associations of brain structural and glucose metabolism covariance networks with memory performance across the cognitive stages of CN, MCI, and probable AD in individuals stratified by Aβ and tau pathologies. Rather than assuming a constant brain-memory association, we demonstrated that brain structural and metabolic network integrity had non-linear associations with memory performance across different cognitive stages; such trajectories exhibited opposing patterns in individuals with and without amyloid pathology. A lower HIP structural network

score was associated with a lower ADNI-mem score and among individuals with amyloid pathology, the strength of this relationship was greatest in early CN and decreased in subsequent cognitive stages. In contrast, the strength of this association was lower and the trajectory was opposite in those with both tau-only and non-amyloid/non-tau pathology. An association between the breakdown of the default mode metabolic network seeded at the ANG with memory deficit was also observed in individuals with amyloid pathology, with the strength of this association peaking in early CN and decreasing gradually before rebounding in the late MCI/dementia stage. Our findings support the AD biomarker hypothetical models by characterising the non-linear influence of brain structural and metabolic networks on memory function across the AD continuum, hence paving the way for early interventions and stage-dependent remedies to modify disease trajectory and improve clinical outcomes.

## Early influence of hippocampal structural network deterioration on memory impairment in asymptomatic amyloid-positive individuals

The HIP structural network is identified to be associated with memory impairment in all three pathology groups which is consistent with the role that the hippocampus plays in memory cognitive domain (*Tulving and Markowitsch, 1998*; *Eichenbaum, 2004*). However, the peak influence of the HIP structural integrity on memory differed among the three pathology groups. The early peak of the association at the CN stage in the amyloid pathology group suggests an early influence of the hippocampal structural network integrity on memory performance in the preclinical AD stage. Our findings are in line with a recent study that compared MRI brain structure models of normal and AD participants across the entire lifespan, showing that the AD model for hippocampus diverged early from normal aging trajectory (*Coupé et al., 2019*). Accumulating evidence also suggests hippocampal volume and thickness as early imaging correlates of verbal memory in preclinical AD (*Bayram et al., 2018*). Furthermore, in a cohort of CN individuals, decreased CSF Aβ42 was associated with hippocampal loss and poorer performance on episodic memory (*Wang et al., 2015*), while an early effect of Aβ on memory mediated by hippocampal atrophy has been demonstrated in non-demented older individuals (*Mormino et al., 2009*; *Lim et al., 2015*; *Mattsson et al., 2015*). These evidence supports our findings of the early influence of structural covariance breakdown in the hippocampal networks on memory performance in the preclinical AD stage.

In our cohort with amyloid pathology, the strength of the association between HIP structural network and memory gradually decreased in the MCI and dementia stages. This suggests that the HIP structural network integrity plays a lesser role on memory performance as the cognitive stages progress. Given that memory impairment is expected to worsen as the cognitive stage progresses, we postulate that structural networks outside the hippocampal/temporal lobes may be increasingly affected while the influence from the hippocampal-based memory network decreases. Indeed, the hippocampus system is well connected to various cortical brain regions in processing memory information (*Treves and Rolls, 1994*) and together with brain structures such as the prefrontal cortex make up a large-scale network to support encoding and retrieval of episodic memory (*Blumenfeld and Ranganath, 2007*). While the medial temporal lobe is well known to be affected early on in the AD process, grey matter regions outside the medial temporal lobes are gradually implicated as the disease progresses to MCI and dementia (*Bayram et al., 2018*). Atrophy in brain regions within the DMN such as the precuneus and the posterior cingulate gyrus are shown to be associated with episodic memory impairment (*Doré et al., 2013*) and decreased inferior frontal gyrus volume is associated with verbal memory decline in MCI patients who converted to AD over time (*Defrancesco et al., 2014*).

## Angular gyrus-seeded default mode network metabolic deterioration plays a key role in memory deficit in the asymptomatic and dementia stages of AD

While impaired glucose uptake in the ANG is consistently shown to be an important feature for predicting memory and executive functioning performance in the later stages of AD (*Jeong et al., 2017*; *Hammond et al., 2020*), our present findings provide further insights into the early critical role of ANG-based metabolic covariance network for intact memory (i.e. earlier peak of beta) in the preclinical AD stage. The ANG, located in the posterior part of the inferior parietal lobule, is one of the major connector hubs that links different subsystems such as the DMN (*Greicius et al., 2004*;

*Tomasi and Volkow, 2011*) that are affected by AD pathophysiology, and is involved in verbal working memory (*Jonides et al., 1998*; *Seghier, 2013*) and episodic memory retrieval (*Ciaramelli et al., 2008*). The role of ANG in memory performance is also implicated by its strong connectivity with the hippocampal system (*Seghier, 2013*) that is critical in episodic and declarative memory functions *Tulving and Markowitsch, 1998*. Furthermore, a recent study showed that Aβ aggregation within the brain's DMN is associated with regional hypometabolism in distant but functionally connected brain regions, including the inferior parietal cortices where the ANG is located (*Pascoal et al., 2019*). Therefore, early malfunctioning of the ANG, as indicated by aberrant metabolic network patterns in our study, may predispose CN individuals with amyloid pathology to a more vulnerable memory system.

Interestingly, we observed that the relationship between ANG-based metabolic covariance network and memory performance gradually decreased in the late CN and MCI stages before increasing in the dementia stage. We postulate that this may represent a metabolic compensatory mechanism in the MCI stage as a manifestation of cognitive reserve to preserve memory function, which has been proposed in AD functional connectivity (FC) studies. Among amnestic MCI individuals, increased FC compared to controls was found within the DMN and between DMN and brain networks such as the frontoparietal control and dorsal attention networks. These abnormal increased FC patterns are associated with lower cognitive performance which suggest a maladaptive compensatory mechanism in the MCI stage (*Gardini et al., 2015*; *Liang et al., 2020*). Similarly, higher nodal topological properties such as the nodal strength, nodal global efficiency and nodal local efficiency, and increased local and medium-range connectivity located in the DMN-related brain regions were also shown in the earlier subjective cognitive decline stage of AD relative to healthy controls (*Chen et al., 2020*). While these evidence supports our hypothesis of a metabolic compensatory mechanism in the late CN/MCI stage of AD, our findings will need to be confirmed in a larger cohort with longitudinal follow-up.

## Modest influence of hippocampal structural network deterioration on memory impairment in individuals with non-amyloid pathology

The strength of the association between HIP structural network covariance and memory function was overall lower in non-AD groups compared to amyloid pathology group, which suggested that the hippocampal network integrity had a more modest influence on memory in individuals without Aβ pathology compared to those with Aβ pathology. In line with our finding, a recent study on 531 deceased older community adults showed that neuropathologies such as AD, cerebrovascular disease and hippocampal sclerosis accounted for 42.6% of the variation in global cognitive decline, whereas hippocampal volume alone only accounted for an additional 5.4% of this variation (*Dawe et al., 2020*). Furthermore, we demonstrated a non-linear and opposing trajectory of this association as the cognitive stage progresses in non-AD groups compared to AD group. Although prior studies have consistently demonstrated that hippocampal atrophy is associated with memory deficits even before the presence of dementia and can predict dementia progression (*Ferrarini et al., 2014*), emerging evidence suggests that the relationship between hippocampal atrophy and memory is also dependent on other factors such as age and cognitive reserve (*Gorbach et al., 2017*; *Vuoksimaa et al., 2013*; *Svenningsson et al., 2019*). Specifically, the association between episodic-memory decline and atrophy in the hippocampus over time was stronger in older than in the middle-aged participants (*Gorbach et al., 2017*). In middle age, hippocampal volume was related to memory in those with low cognitive reserve, but not in those with high cognitive reserve (*Vuoksimaa et al., 2013*). Excitingly, our findings shed new insights that the associations of memory decline with both hippocampal structural network integrity and years of education (i.e. a proxy for cognitive reserve) were also dependent on the presence/absence of amyloid pathology and the level of cognitive impairment.

## Strengths and limitations

The main strength of the present study is the inclusion of individuals from the ADNI cohort with well characterised neuropsychological, multimodal neuroimaging, and AD biomarker data. This enables the study of the relationships between metabolic, structural brain networks, and memory performance specifically in individuals within the AD continuum and those without amyloid pathology. Nevertheless, there are a few limitations in our study. First, the ADNI cohort consists of self-selected individuals participating in a study focusing on AD research, e.g., relatively more amyloid positive individuals, which may introduce selection bias and limit the generalisability of our findings to a broader

community. Second, our study design is cross-sectional thus provides only indirect evidence on the underlying brain-behaviour relationship. Therefore, a larger population-based longitudinal study is needed to characterise within-subject trajectories of brain-behaviour relationships across the disease continuum. Third, while we characterised the amyloid and tau status of our cohort using CSF amyloid and p-tau, we did not consider the spatial patterns of amyloid and tau brain deposition. Further studies are needed to elucidate the complex spatial and temporal trajectories of structural and meta-bolic networks in the various non-amyloid tauopathies and how the presence of amyloid affects the tau-metabolism-memory associations across the disease continuum. Fourth, we estimated the indi-vidual brain network scores based the group-level salience map derived from all the participants, which could potentially be sensitised to the relative imbalance of group sizes across diagnoses and/or A/T categories. Nevertheless, we obtained similar findings when using group-level salience maps that were generated from CN individuals only, which indicated the robustness of our findings. Moving forward, a large independent cohort of CN individuals with minimum amyloid and tau pathology will be a better reference (*Liu et al., 2021*). Last, a multiplex graph-based approach can be applied to quantify differential network contributions to memory in the future studies (*Canal-Garcia et al., 2022*).

In conclusion, our findings support the AD hypothetical models that the association between neurodegeneration and memory dysfunction is non-linear across cognitive stages and depends on the type of pathology. The early influence of metabolic and structural covariance breakdown in the default mode and hippocampal networks on memory performance underscore the importance of early intervention in preclinical AD.

## Materials and methods

### Participants

Data used in this article were obtained from the ADNI database (https://adni.loni.usc.edu/). The ADNI was launched in 2003 as a public-private partnership, led by Principal Investigator Michael W. Weiner, MD. The primary goal of ADNI has been to test whether serial MRI, PET, other biological markers, and clinical and neuropsychological assessment can be combined to measure the progression of MCI and early AD.

In this study, we first selected 812 participants to define seed regions for brain network derivation (*Figure 1*, step 1). All of the images passed the visual quality control. Among them, 232 were CN, 413 were MCI, and 167 were probable AD. We then identified 708 participants (610 from ADNI-2 and 98 from ADNI-GO) from the above cohort who underwent neuropsychological assessments, and lumbar puncture in addition to [18 F]FDG PET and 3T T1-weighted MRI scans to form the main study cohort (*Figure 1*, steps 2 and 3). Among them, 195 were CN, 374 were MCI and 139 were probable AD (*Table 1*). A larger validation dataset was created for replication by including another 468 individuals (377 from ADNI-1, 38 from ADNI-2, and 53 from ADNI-GO) who underwent 1.5T T1-weighted MRI scan (refer as validation dataset 1). We also performed an additional validation analysis on the fully independent 468 participants (see *Figure 1—figure supplement 1* flowchart at right; refer as valida-tion dataset 2). *Figure 1—figure supplement 1* showed the flowchart of study participant selection.

Following ADNI diagnostic criteria (*Petersen et al., 2010*), we defined CN as those with mini-mental state examination (MMSE) scores ≥24 and clinical dementia rating (CDR) 0, and showed no signs of depression, MCI, or dementia. MCI was defined as those with MMSE scores ≥24 and CDR 0.5, subjective and objective memory loss, absence of significant levels of impairment in other cognitive domains, essentially preserved activities of daily living, and an absence of dementia. Probable AD was defined as those with MMSE scores ≤26, CDR ≥0.5 and meeting the NINCDS/ADRDA criteria for probable AD.

Aβ (A) and tau (T) pathologies were measured using CSF $A\beta_{1-42}$ and CSF p-tau$_{181p}$. More details were in Supplementary methods. Using the ADNI published cutoffs of $A\beta_{1-42}$<192 pg/mL and CSF p-tau$_{181p}$ >23 pg/mL to define the presence of Aβ and tau pathology, respectively (*Shaw et al., 2009*), the main study cohort was further stratified into three pathology groups: A-T- (non-amyloid/non-tau), A-T+ (tau only) and A+T-/A+T + (amyloid pathology; *Table 1*). There was no significant difference in age, gender, years of education, and *APOE* ε4 status among CN, MCI, and probable AD individuals in the A-T- and A-T +groups (*Table 1*). The proportion of *APOE* ε4 carriers was lower in CN compared to MCI and dementia individuals in the A+T-/A+T + group.

The ADNI study was approved by the Institutional Review Boards of all of the participating institutions and informed written consent was obtained from all participants at each site.

### Neuropsychological assessment

The ADNI-mem is a validated composite memory score derived using data from the ADNI neuropsychological battery (*Crane et al., 2012*). More details were in Supplementary methods.

### Image acquisition and preprocessing

All participants from the main dataset underwent T1-weighted MRI scans according to the standardised ADNI protocol using 3-Tesla scanners. Additional participants who underwent structural MRI brain scans using 1.5-Tesla scanners were included for validation of the findings. All participants also underwent [$^{18}$F]FDG PET to study cerebral glucose metabolism (185 MBq (5 mCi), dynamic 3D scan of six 5 min frames 30–60 min postinjection).

All T1-weighted MRI scans were corrected for field distortions and processed using the CIVET image processing pipeline (https://www.bic.mni.mcgill.ca/ServicesSoftware/CIVET) to generate the GM probability maps as previously described (*Kang et al., 2021*). [$^{18}$F]FDG PET images were processed with an in-house processing pipeline as described in our previous work (*Nilson et al., 2017*). Further details on image parameters and preprocessing were in Supplementary methods.

### Statistical analyses

Between-group differences in demographic characteristics and clinical assessments were tested among CN, MCI, and probable AD groups. Either a one-way ANOVA or a chi-squared test was used depending on the nature of the variable.

### Seed definition: group comparison on GMV and glucose metabolic pattern between CN and probable AD

As shown in *Figure 1* (step 1), the 12 seed coordinates from the DMN, the SN, the ECN and the memory network were determined based on the group comparisons of the GMV probability and glucose metabolic spatial maps between CN and probable AD individuals using a permutation test (randomise, FSL, 5000 permutations). Effects of age, gender, years of education, and APOE ε4 genotype were regressed out. The field strength (i.e. 1.5T or 3T) was included as an additional covariate when the tests were performed using the validation dataset 1 (*Supplementary file 1*). The resulting GMV and metabolic group difference maps (i.e. CN greater than probable AD) were thresholded using threshold-free cluster enhancement with an alpha level of 0.05 (corrected at family-wise error [FWE] rate). We superimposed the two thresholded t statistical maps (GMV and metabolic) and summed the t-scores at each voxel. Spherical seeds (with 4 mm radius) were then defined based on the peak foci of the above network key regions showing atrophy and hypometabolism in probable AD compared to CN (*Supplementary file 4*).

### Brain metabolic and structural network derivation: seed PLS analyses

We used seed PLS to identify covariance patterns between GMV/metabolism in each seed region and those of all other voxels in the whole brain (*Figure 1*, step 2). The seed value was defined as the average GMV/metabolism values within each predefined seed from step 1. For each seed region, the vector **Y** representing the seed values concatenated across all the participants was cross-correlated with a matrix **X,** representing the vectorised whole-brain GMV (or metabolism) images of all the participants. Both the seed vector **Y** and the image matrix **X** were centered and normalised such that the vector of correlations **R** was computed as:

$$\mathbf{R} = \mathbf{Y^T} \cdot \mathbf{X}$$

Using singular value decomposition, the correlation vector **R** was decomposed into a set of mutually orthogonal latent variables (LVs) comprising three matrices:

$$\mathbf{R} = \mathbf{v} \cdot \mathbf{s} \cdot \mathbf{u^T}$$

where **s** is the diagonal matrix of singular values, and **v** and **u** are the orthonormal matrices of left and right singular vectors, which are also called saliences in the PLS terminology. The left and right singular vectors respectively represent the seed profiles and the whole-brain patterns that best characterise the correlation vector **R**. Therefore, the brain salience **u** captures the brain covariance or network pattern that is of interest. The number of LVs derived is equal to the rank of the correlations vector **R**. The LVs were tested for statistical significance with 1000 permutations. The stability of each voxel in the brain salience of the LV was validated using a bootstrap ratio, calculated by dividing the voxel salience value by its standard error, estimated by bootstrapping (500 times).

The resulting significant LV from the PLS analyses of each of the 12 seeds (all p<0.0001) corresponded to reliable patterns of structural or metabolic covariance network associated with that seed (see *Figure 2A*, *Figure 3A*, *Figure 2—figure supplement 1A* and *Figure 3—figure supplement 1A*).

To represent individual-level brain salience maps of the identified LV for each seed PLS model, the original matrix **X** was projected onto the brain salience **u** (representing the network map), which was computed by:

$$\mathbf{L_X} = \mathbf{X} \cdot \mathbf{u}$$

where $\mathbf{L_X}$ is a vector of brain structural or metabolic network scores across all the participants.

We calculated the brain network score for each of the 12 networks in both FDG and GMV modalities separately. For HIP, ANG, INS, PPC, and DLPFC, we averaged the left and right brain network scores. In total, each participant had 14 brain network scores (i.e. two for each of the 7 seed regions, including HIP, ANG, PCC, mPFC, INS, PPC, and DLPFC), which reflect structural or metabolic network pattern expression.

## Stage- and pathology-dependent associations between brain networks and memory impairment: SVC modelling

With ADNI-mem as the dependent variable, the SVC models have the following form:

$$y_i(t_k) = \sum_{j=1}^{p} \beta_j(t_k) x_{ij}(t_k) + \varepsilon_i(t_k)$$

where the dependent variable $y_i(t_k)$ represents the cognitive score for subject i $(i = 1, 2, \ldots, n)$ at the bin $t_k$ $(k = 1, 2, \ldots, K)$, $x_{ij}(t_k)$ is the $j^{th}$ $(j = 1, 2, \ldots, p)$ predictor (FDG/GMV network scores and nuisance variables; see below) of subject i at the bin $t_k$. Both the dependent variable and all predictors were standardised to z-scores within each pathology group. $\beta_j(t_k)$ is the coefficient function depending on bin $t_k$ for each feature $j$ and $\varepsilon_i(t_k)$ is the independent and identically distributed random errors at $t_k$. The coefficient function $\beta_j(t_k)$ is approximated using linear combinations of a set of B-spline basis. To simultaneously achieve regression model fitting and variable selection, the LASSO; *Tibshirani, 1996* is applied to estimate $\beta_j(t_k)$ by minimising the following penalised least squares function:

$$\frac{1}{2n} \sum_{i=1}^{n} \sum_{k=1}^{K} \left[ y_i(t_k) - \sum_{j=1}^{p} \beta_j(t_k) x_{ij}(t_k) \right]^2 + \lambda \sum_{j=1}^{p} \sqrt{\int \beta_j^2(t) \, dt}$$

where $\lambda$ is the sparse penalty tuning parameter, which was chosen by a fivefold cross-validation method.

We ran each SVC model for 100 repetitions and reported the brain measures that were consistently selected by more than 90 repetitions. These measures were interpreted as a set of critical brain GMV/metabolism networks that contributed to memory across the cognitive stages, with a vector of beta coefficients reflecting stage-dependent (non)linearity in the network-memory association.

To assess the stability of these beta coefficients, we calculated the mean and standard error of the stage/pathology-dependent coefficients estimated from all 100 repetitions. Moreover, to assess the specificity of the selected networks, we randomly permuted the memory scores 100 times across the participants and repeated the SVC modelling 100 times within each of the 100 permuted data sets, following our previous approach (*Hong et al., 2015*). These 'null' permutations should yield

inconsistent selection of predictors, if any, as compared to our actual models. SVC modelling was performed by in-house R scripts based on Daye and colleagues (*Daye et al., 2012*).

To further confirm that our findings were robust, we repeated the analyses with another ordering strategy which did not divide MCI and probable AD into two separate groups (i.e. CDR-SOB and age ordering were done across all individuals with either MCI or probable AD diagnosis) in each pathology group (*Figure 1—figure supplement 2B*).

To compare the brain metabolic and structural network scores between different cognitive stages and different pathology groups (*Figure 1*, step 3), we performed separate non-parametric one-way ANOVA analyses (5000 permutations, alpha = 0.05) to test whether there was group differences across cognitive stages and across different pathology groups for each network, followed by posthoc non-parametric two-sample t-tests (5000 permutations, alpha = 0.05). We performed Bonferroni-Holm correction for the three pair's t-tests (adjusted alpha = 0.05). The nuisance variables including age, gender, education years, *APOE* ε4, ICV, and scan site were regressed out from the network scores before non-parametric ANOVA.

## Acknowledgements

Data used in preparation of this article were obtained from the Alzheimer's Disease Neuroimaging Initiative (ADNI) database (adni.loni.usc.edu). As such, the investigators within the ADNI contributed to the design and implementation of ADNI and/or provided data but did not participate in analysis or writing of this report. A complete listing of ADNI investigators can be found at http://adni.loni.usc.edu/wp-content/uploads/how_to_apply/ADNI_Acknowledgement_List.pdf.

Data collection and sharing for this project was funded by the Alzheimer's Disease Neuroimaging Initiative (ADNI) (National Institutes of Health Grant U01 AG024904) and DOD ADNI (Department of Defense award number W81XWH-12-2-0012). ADNI is funded by the National Institute on Aging, the National Institute of Biomedical Imaging and Bioengineering, and through generous contributions from the following: AbbVie, Alzheimer's Association; Alzheimer's Drug Discovery Foundation; Araclon Biotech; BioClinica, Inc.Inc; Biogen; Bristol-Myers Squibb Company; CereSpir, Inc.Inc; Cogstate; Eisai Inc.Inc; Elan Pharmaceuticals, Inc.Inc; Eli Lilly and Company; EuroImmun; F Hoffmann-La Roche Ltd and its affiliated company Genentech, Inc.Inc; Fujirebio; GE Healthcare; IXICO Ltd.; Janssen Alzheimer Immunotherapy Research & Development, LLC.; Johnson & Johnson Pharmaceutical Research & Development LLC.; Lumosity; Lundbeck; Merck & Co., Inc.Inc; Meso Scale Diagnostics, LLC.; NeuroRx Research; Neurotrack Technologies; Novartis Pharmaceuticals Corporation; Pfizer Inc.Inc; Piramal Imaging; Servier; Takeda Pharmaceutical Company; and Transition Therapeutics. The Canadian Institutes of Health Research is providing funds to support ADNI clinical sites in Canada. Private sector contributions are facilitated by the Foundation for the National Institutes of Health (https://www.fnih.org/). The grantee organization is the Northern California Institute for Research and Education, and the study is coordinated by the Alzheimer's Therapeutic Research Institute at the University of Southern California. ADNI data are disseminated by the Laboratory for Neuro Imaging at the University of Southern California. We also acknowledge the funding support from Yong Loo Lin School of Medicine, National University of Singapore (JHZ), the Duke-NUS/Khoo Bridge Funding Award (JHZ, KBrFA/2019–0020), NMRC Open Fund Large Collaborative Grant (JHZ, OFLCG09May0035) and NMRC New Investigator Grant (KPN, MOH-CNIG18may-0003).

## Additional information

### Competing interests

Serge Gauthier: received consulting fees from CERVEAU Therapeutics, Biogen Canada, Roche Canada, TauRx, honoraria from Biogen Canada, and payment for participation on the DIAN-TU Washington University drug selection committee. Nagaendran Kandiah: received grants from Novartis Pharmaceuticals and Schwabe Pharmaceuticals, honoraria and support (for travel and/or meetings) from Eisai Pharmaceuticals, Novartis, Schwabe and Lundbeck, and participated on the Asian Society Against Dementia committee and Vascog Asia. Juan Helen Zhou: Reviewing editor, eLife. Alzheimer's Disease Neuroimaging Initiative: The other authors declare that no competing interests exist.

## Funding

| Funder | Grant reference number | Author |
|---|---|---|
| Duke-NUS Medical School | Duke-NUS/Khoo Bridge Funding Award (KBrFA/2019-0020) | Juan Helen Zhou |
| National Medical Research Council | NMRC Open Fund Large Collaborative Grant (OFLCG09May0035) | Juan Helen Zhou |
| National Medical Research Council | NMRC New Investigator Grant (MOH-CNIG18may-0003) | Kok Pin Ng |
| Yong Loo Lin School of Medicine Research funding | | Juan Helen Zhou |

The funders had no role in study design, data collection and interpretation, or the decision to submit the work for publication.

## Author contributions

Kok Pin Ng, Conceptualization, Data curation, Supervision, Methodology, Writing – original draft, Project administration, Writing – review and editing; Xing Qian, Conceptualization, Data curation, Software, Formal analysis, Validation, Visualization, Methodology, Writing – original draft, Writing – review and editing; Kwun Kei Ng, Fang Ji, Software, Methodology, Writing – review and editing; Pedro Rosa-Neto, Serge Gauthier, Nagaendran Kandiah, Supervision, Project administration, Writing – review and editing; Juan Helen Zhou, Conceptualization, Resources, Supervision, Funding acquisition, Methodology, Writing – review and editing; Alzheimer's Disease Neuroimaging Initiative, Data curation, Funding acquisition, Resources

## Author ORCIDs

Kok Pin Ng http://orcid.org/0000-0003-3787-8944
Xing Qian http://orcid.org/0000-0002-6653-7121
Kwun Kei Ng http://orcid.org/0000-0002-0584-7679
Pedro Rosa-Neto http://orcid.org/0000-0001-9116-1376
Juan Helen Zhou http://orcid.org/0000-0002-0180-8648

## Ethics

Human subjects: Data used in the preparation of this article were obtained from the Alzheimer's Disease Neuroimaging Initiative (ADNI) database (adni.loni.usc.edu). The ADNI was launched in 2003 as a public-private partnership, led by Principal Investigator Michael W. Weiner, MD. The primary goal of ADNI has been to test whether serial magnetic resonance imaging (MRI), positron emission tomography (PET), other biological markers, and clinical and neuropsychological assessment can be combined to measure the progression of mild cognitive impairment (MCI) and early Alzheimer's disease (AD). The ADNI study was approved by the Institutional Review Boards of all of the participating institutions and informed written consent was obtained from all participants at each site.

## Decision letter and Author response

Decision letter https://doi.org/10.7554/eLife.77745.sa1
Author response https://doi.org/10.7554/eLife.77745.sa2

---

# Additional files

## Supplementary files

• Supplementary file 1. Participants demographics for network seed definition step.

• Supplementary file 2. Study participant demographics of the validation dataset 1 for the PLS-SVC model.

• Supplementary file 3. Study participant demographics of the validation dataset 2 for the SVC model.

• Supplementary file 4. The coordinates of the peak foci of regions showing difference in metabolism and grey matter volume between probable AD and healthy controls.

- Transparent reporting form

## Data availability

ADNI data used in this manuscript are publicly available at adni.loni.usc.edu, subject to adherence to the ADNI Data Use Agreement and publications' policies (https://ida.loni.usc.edu/collaboration/access/appLicense.jsp). Guidelines to apply for data access can be found in https://adni.loni.usc.edu/data-samples/access-data/#access_data. Codes used in this manuscript are available at https://github.com/hzlab/2021Qian_ADNI_FDG, (copy archived at swh:1:rev:f59dfbc9520d6bf49bfe5345b3d6d72ddddc4187).

The following previously published dataset was used:

| Author(s) | Year | Dataset title | Dataset URL | Database and Identifier |
|---|---|---|---|---|
| Aisen PS, Petersen RC, Beckett LA, Donohue MC, Gamst AC, Harvey DJ, Jack CR, Jagust WJ, Shaw LM, Toga AW, Trojanowaki JQ, Weiner MW | 2010 | Alzheimer's Disease Neuroimaging Initiative (ADNI) | https://adni.loni.usc.edu/ | loni, ADNI |

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

## Appendix 1

### Supplementary methods

#### CSF analysis

CSF AD biomarkers of Aβ1-42 and CSF p-tau181p were measured using the Luminex multiplex platform (Luminex, Austin, TX, USA) and Innogenetics INNO-BIA AlzBio3 (Innogenetics, Ghent, Belgium) immunoassay reagents. The details of the ADNI methods for the acquisition and measurement of CSF can be found at https://adni.loni.usc.edu/.

#### Neuropsychological assessment

The ADNI-mem is a validated composite memory score derived using data from the ADNI neuropsychological battery (*Crane et al., 2012*). A modern psychometric approach was used to analyse the Rey Auditory Verbal Learning Test, AD assessment schedule-cognition (ADAS-cog), MMSE, and Logical Memory tests to obtain a composite memory score. In ADNI-mem composite scores, lower scores reflect poorer memory performance. The details of the ADNI protocols for the neuropsychological assessments and the methods for developing the ADNI-mem can be found at https://adni.loni.usc.edu/.

#### Image acquisition and preprocessing

All participants from the main dataset underwent T1-weighted MRI scans according to the standardised ADNI protocol using 3-Tesla GE, Philips, and Siemens MRI scanners with a sagittal volumetric magnetisation-prepare rapid-acquisition gradient echo (MPRAGE) sequence (TR = 2300ms, TE = minimum full, approximate TI = 900ms, Slice Thickness = 1.2, flip-angle=9°) or T1-weighted accelerated sagittal inversion-recovery spoiled gradient-recalled (SPGR) sequence (TR = 400ms, TE = minimum full, flip-angle=11°, slice thickness = 1.2 mm and FOV = 26 cm). Additional participants who underwent structural MRI brain scans using 1.5-tesla GE, Philips, and Siemens MRI scanners were included for validation analyses. For these participants, T1-weighted MRI scans were acquired using an MPRAGE sequence with TR = 2400ms, minimum full TE, TI = 1000ms, Slice thickness = 1.2, and flip angle of 8° (scan parameters vary between sites, scanner platforms, and software versions).

All participants also underwent [$^{18}$F]FDG PET to study cerebral glucose metabolism (185 MBq [5 mCi], dynamic 3D scan of six 5 min frames 30–60 min postinjection). Further details on MRI and PET acquisition parameters can be found at the ADNI website http://adni.loni.usc.edu/methods.

#### Voxel-based morphometry

All T1-weighted MRI scans were corrected for field distortions and processed using the CIVET image processing pipeline (https://www.bic.mni.mcgill.ca/ServicesSoftware/CIVET). The MRI images underwent non-uniformity correction, brain masking and segmentation, and normalisation to the Montreal Neurological Institute (MNI) space with affine and non-linear transformation. An in-house processing pipeline based on MINC toolkits was then applied to generate voxel-based morphometry (VBM) images based on the CIVET outputs as previously described (*Kang et al., 2021*). In brief, a log Jacobian determinant was derived based on the non-linear vector field from the CIVET outputs, followed by transformation into a scalar, modulated with grey matter probability mask. The GM probability maps were then smoothed with an 8 mm Full-Width at Half-Maximum (FWHM) Gaussian kernel.

#### [$^{18}$F]FDG PET processing

[$^{18}$F]FDG PET images were processed with an in-house processing pipeline as described in our previous work (*Ng et al., 2017*). The preprocessed images from the ADNI database were smoothed with an 8 mm FWHM Gaussian kernel, followed by linear co-registration and non-linear spatial normalisation to the MNI 152 standardised space with the use of transformation matrices derived from the PET native to MRI native space and the MRI native to the MNI 152 space. The voxel-wise brain glucose metabolism standardised uptake value ratio (SUVR) maps were then generated with the pons as the reference region.

