## [Editor Report]

This paper presents important information about how potential network-based structural and metabolic imaging biomarkers are associated with memory performance during distinct disease stages, in line with previous hypothetical biomarker models. The study is conceptually sound and methodologically convincing and will be of interest to neuroscientists and medical professionals involved in the study of Alzheimer's disease and related neurodegenerative conditions.

---

## [Decision Letter]

**Decision letter after peer review:**

Thank you for submitting your article "Stage dependent differential influence of metabolic and structural networks on memory across Alzheimer's disease continuum" for consideration by *eLife*. Your article has been reviewed by 2 peer reviewers, and the evaluation has been overseen by a Reviewing Editor and Jeannie Chin as the Senior Editor. We thank you so much for your patience during this unusually long review period. The following individual involved in review of your submission has agreed to reveal their identity: Amy Kuceyeski (Reviewer #2).

In your revision, as you address the Reviewers' points, please consider these Essential Revisions:

1) Please clarify whether the validation cohort was truly independent from the original analyses, and include additional information on how the analyses were done.

2) Add discussion regarding the analysis method and whether it may be sensitized to the relative imbalance of group sizes across diagnoses or other parameters or outliers.

3) Please demonstrate that the variance is similar across diagnostic groups, or use a non-parametric test. Also, clarify whether p-values are corrected for multiple comparisons in all figures.

4) It would be helpful to include an assessment of metabolic network scores for a region that might not be (or might be least) affected in AD as a negative control, to address the question of whether the effects being measured are global or localized effects.

5) Please include an analysis that uses a method other than LASSO to assess correlations between network measures.

*Reviewer #1 (Recommendations for the authors):*

– I would argue that for the PLS analysis, an approach similar to Liu et al. 2019 Molecular Psychiatry would be more reasonable, as back-projecting to a template salience map derived from a healthy subgroup would mostly eliminate the concerns I mention in the public review. Otherwise, a clear justification for the employed scheme should be included.

– The bar charts depicting the network scores suggest that the variance is unequal across diagnostic groups, which would violate ANOVA assumptions. It would increase the strength of the group comparison results if a non-parametric test could be applied.

– It might be outside the scope of this paper but it would be interesting to see a quantification of the divergence between metabolic and structural network contributions to memory scores, e.g. by applying a multiplex graph-based approach (as in, for example, Canal-Garcia et al. 2022 Cerebral Cortex).

*Reviewer #2 (Recommendations for the authors):*

In figure 2, please represent the data in panel B as a violin or raincloud plot – the bar chart as it is shown obscures the details of the data distribution.

More details are needed in the main text to understand how the z-scores in Figures 3B and 4B are calculated. What value of the GMV and metabolic maps from the individual are used to derive the individual scores? Raw GMV per voxel within the GM covariance network?

Once the network scores are found, what group is used to z-score? From looking at the bar charts, it seems that the entire 708 subjects group was used to calculate the z-scores? or CN only?

From looking at the widespread nature of the metabolic networks in Figure 2A (and to a lesser extend the structural networks in Figure 3A) and the near constant values in the bar plots within the patient groups, it appears that the 7 "different" networks may be measuring more of a global effect rather than a localized one. It would be good to see metabolic network scores for a region that is hypothesized to be least effected in AD as a negative control. If this network still has the same overall effect, then maybe the pathological network effect being measured here is a global phenomena?

It seems odd that the CN group would have lower structural network scores than the MCI in the A-T- group. Please discuss this somewhat surprising finding.

There are many comparisons being done here (although it is unclear how many) – are p-values being corrected for multiple comparisons in Figures 2B and 3B? Please add these details.

There is likely a lot of correlation between the network measures (as evidenced by the fact that the network scores are very consistent across regions within the same group). It is known that LASSO will randomly suppress one of two correlated variables. Please use a different penalty (perhaps ridge) that does not have this drawback so that the interpretation of the coefficients can be more trustworthy.

Relatedly, Sup Figure 9 showing the variable selection frequency for permuted datasets is a bit confusing – the frequency ranges are 0 to 5 (unclear what unit this is representing), do not appear to be uniform/random and vary greatly between the main and validation dataset. I am not sure how these results support the main conclusions about the SVC model. Replication of the main findings using something other than Lasso/Elastic Net would be helpful to support the main findings.

The replication study is a bit confusing. It appears that 468 individuals were added to the original set of 812 to obtain a set of 1280 individuals, 859 of which were used in the final step of the analysis. Unless there is a misunderstanding, doesn't this mean that the "replication dataset" is actually mostly the original dataset? it would be stronger to replicate the study using only the 468 independent individuals – otherwise it is not a replication study.

---

## [Author Response]

In your revision, as you address the Reviewers' points, please consider these Essential Revisions:1) Please clarify whether the validation cohort was truly independent from the original analyses, and include additional information on how the analyses were done.

Clarification on the validation cohort and findings is discussed in our detailed response to Reviewer #2, comment #9. Briefly, the validation dataset 1 was not fully independent from the main dataset due to limited tau and amyloid information. As suggested by Reviewer #2, we have performed a separate validation analysis using only data from the 468 individuals (i.e., validation dataset 2) that were truly independent from the main dataset. The sparse varying coefficient (SVC) modelling showed that results in the A+T-/A+T+ group were consistent with those from the main dataset. In other words, we replicated the key findings in two independent datasets (main dataset and validation dataset 2) that in amyloid pathology group, there was an early influence of hippocampal structural network deterioration on memory impairment in the preclinical stage, and a biphasic influence of the angular gyrus-seeded default mode network metabolism on memory in both preclinical and dementia stages.

2) Add discussion regarding the analysis method and whether it may be sensitized to the relative imbalance of group sizes across diagnoses or other parameters or outliers.

We have added this point in *Discussion section* (also see our detailed response to Reviewer #1, comment #1). As suggested by Reviewer #1, we performed an additional validation analysis that used only cognitively normal (CN) individuals from validation dataset 1 to derive the brain salience maps in the partial least square analyses. We then calculated the individual brain network scores based on these CN derived brain salience maps and repeated our SVC modelling using these new set of brain network scores. The results were consistent with our main findings in which we derived the brain salience maps from participants of all cognitive stages.

3) Please demonstrate that the variance is similar across diagnostic groups, or use a non-parametric test. Also, clarify whether p-values are corrected for multiple comparisons in all figures.

As suggested, we have now used non-parametric tests and performed multiple comparisons corrections for the group comparisons in the revised manuscript. More details could be found in the responses to Reviewer #1, comment #2 and Reviewer #2, comment #6.

4) It would be helpful to include an assessment of metabolic network scores for a region that might not be (or might be least) affected in AD as a negative control, to address the question of whether the effects being measured are global or localized effects.

To address this point, we have clarified our rationale of defining the AD-related seeds based on the grey matter probability and glucose metabolic contrast maps between controls and AD patients. Moreover, as suggested, we performed a new set of analyses based on sensorimotor metabolic network scores derived from primary sensorimotor region, which is usually less affected in the early stages of AD. Unlike those networks defined using the seeds showing difference between AD and controls, sensorimotor metabolic network showed no differences between the pathology groups in all three cognitive stages. This suggests that the observed effects in our original findings are more network-specific rather than global. More details could be found in the response to Reviewer #2, comment #4.

5) Please include an analysis that uses a method other than LASSO to assess correlations between network measures.

As suggested, we have performed an additional analysis that used Ridge penalty instead of LASSO method in the SVC modelling on the main dataset. Using the Ridge penalty method, we found similar disease-dependent associations of hippocampal structural and angular gyrus-based metabolic network scores with memory scores, consistent with our original findings using LASSO. More details could be found in the response to Reviewer #2, comment #7.

Reviewer #1 (Recommendations for the authors):– I would argue that for the PLS analysis, an approach similar to Liu et al. 2019 Molecular Psychiatry would be more reasonable. Otherwise, a clear justification for the employed scheme should be included.

We thank Reviewer #1 for raising this important comment. There are pros and cons in terms of deriving group-level salience maps based on the whole group or only the healthy subgroup. We adopted the former approach following previous work (Spreng, DuPre et al. 2019, Veldsman, Cheng et al. 2020) because we do not want the individual network scores based on the group-level salience maps to be biased by a certain subset of individuals, which may cause confounding effects during group comparisons or SVC modelling across different cognitive stages and pathology groups. The latter approach, i.e., generating templates based on the healthy group, would work better in terms of generating a clean norm if there is a large independent cohort of cognitively normal individuals with minimal amyloid and tau deposition and matched demographics. We agree that the brain salience maps generated from an independent healthy group will be less sensitized to the relative imbalance of group sizes across diagnoses in the testing sample. Unfortunately, we do not have that in the current dataset.

Nevertheless, following the suggestion from Reviewer #1, we have performed additional validation analyses to test if the results remained when the template salience maps are derived from a healthy subgroup. Although we do not have a large independent cohort of cognitively normal (CN) individuals to derive the unbiased brain network templates, we took the less optimal approach, i.e., running the same analysis on the validation dataset 1 which has a larger sample of CN participants. Specifically, we performed PLS analysis on these CN individuals to generate the brain salience maps. We then projected the individual brain maps onto the newly CN-derived group-level salience maps to obtain the individual brain metabolic and structural network scores. Lastly, we performed SVC modelling on the new set of individual network scores.

Overall, our findings remained consistent with the original results using this approach. The hippocampal structural network score and the angular gyrus-based metabolic network score were selected as the key predictors for memory function (see Results pp. 14, paragraph 2, Figure 4—figure supplement 1B and Figure 5—figure supplement 1A). Furthermore, the trajectories of brain-memory associations across the cognitive stages in all three pathology groups were also consistent with the original results.

We have added these new validation results in the revised manuscript (see Replication in the validation dataset in Results, Figure 4—figure supplement 1B and Figure 5—figure supplement 1A) to demonstrate the robustness of our findings. We have also discussed this point in Strengths and Limitations in Discussion section.

– The bar charts depicting the network scores suggest that the variance is unequal across diagnostic groups, which would violate ANOVA assumptions. It would increase the strength of the group comparison results if a non-parametric test could be applied.

We thank Reviewer #1 for this pertinent point. As advised, we have applied non-parametric tests on the group comparisons and the results were similar to the original analyses. We have revised the findings in the current manuscript. We have also revised the *Statistical analyses* sub*-*section under the *Methods* section.

– It might be outside the scope of this paper but it would be interesting to see a quantification of the divergence between metabolic and structural network contributions to memory scores, e.g. by applying a multiplex graph-based approach (as in, for example, Canal-Garcia et al. 2022 Cerebral Cortex).

We thank Reviewer #1 for recommending this paper by Canal-Garcia et al. (Canal-Garcia, Gómez-Ruiz et al. 2022). We agree that this is an excellent idea but given the current SVC model, we were unable to apply this approach directly. Therefore, we have added this paper in our reference list and mentioned the multiplex graph-based approach as one of the future directions under the *Discussion section* of the revised manuscript (pp. 21, paragraph 1).

Reviewer #2 (Recommendations for the authors):1. In figure 2, please represent the data in panel B as a violin or raincloud plot – the bar chart as it is shown obscures the details of the data distribution.

We thank Reviewer #2 for the recommendation. We have revised Figure 2B, Figure 3B, Figure 2—figure supplement 1B and Figure 3—figure supplement 1B by representing the data as half-violin plots.

2. More details are needed in the main text to understand how the z-scores in Figures 3B and 4B are calculated. What value of the GMV and metabolic maps from the individual are used to derive the individual scores? Raw GMV per voxel within the GM covariance network?

We thank Reviewer #2 for this pertinent comment. In the revised Figures 2B and 3B, the z-scores of the brain scores were calculated across all the subjects. We have added this in the figure legends (Figures 2 and 3).

To calculate the individual brain network scores, we projected the original individual whole-brain grey matter probably maps or glucose metabolic maps (represented by the image matrix **X** [number of subjects by number of voxels]) onto each of the brain covariance networks (vectorized whole-brain map [number of voxels by one]). The calculated vector (number of subjects by one) represented brain structural or metabolic network scores across all the participants. Then, we performed SVC modelling for each pathology group separately. The z-scores of the predictors, including the brain scores and the covariates, were calculated within each pathology group (i.e., A-T-, A-T+ and A+T-/A+T+ groups).

We have revised this methodology to improve its clarity in the Statistical analyses sub-section under Methods section in the revised manuscript.

3. Once the network scores are found, what group is used to z-score? From looking at the bar charts, it seems that the entire 708 subjects group was used to calculate the z-scores? or CN only?

We thank Reviewer #2 for this comment. In Figures 2B and 3B, the z-scores of the brain scores were calculated within the entire 708 subjects when performing group comparisons. For SVC modelling within each pathology group, the z-scores of the brain scores were calculated within each pathology group. We have described this methodology to improve its clarity in the Statistical analyses sub-section under Methods section in the revised manuscript as advised.

4. From looking at the widespread nature of the metabolic networks in Figure 2A (and to a lesser extend the structural networks in Figure 3A) and the near constant values in the bar plots within the patient groups, it appears that the 7 "different" networks may be measuring more of a global effect rather than a localized one. It would be good to see metabolic network scores for a region that is hypothesized to be least effected in AD as a negative control. If this network still has the same overall effect, then maybe the pathological network effect being measured here is a global phenomena?

We thank Reviewer #2 for this important comment. We would like to clarify that the definition of our seeds was based on the comparisons of the grey matter volume probability maps and glucose metabolic spatial maps between the probable AD group and CN group. Therefore, the brain networks measured from these AD-related seeds are more likely of a regional than global effect. We agreed that the color contrast of the original figure was confusing. In the revised manuscript, we have revised *Figure 2A* to demonstrate the key regions in each metabolic network.

Moreover, as suggested by Reviewer #2, we defined a seed in the primary motor cortex that is less affected in AD as reported by Elizabeth et al. (DuPre and Spreng 2017). It was also not in the AD and CN contrast maps. Using the same PLS analyses, we then derived the individual metabolic brain scores for the sensorimotor network (see Author response image 1) and compared them across the three pathology groups (see Author response image 1). Using non-parametric one-way ANOVA tests, there were no significant differences in the metabolic brain scores between the pathology groups in all three cognitive stages. This is different from those seeds defined in our original analyses, which exhibited differences between all three pathology groups. In the A+T-/A+T+ pathology group, individuals with probable AD had lower metabolic brain scores than individuals who were CN and MCI. This is consistent with the current understanding of the AD disease process, where the low-level sensorimotor networks are more likely to be affected in late clinical stage, as a result of widespread hypometabolism (Lowe, Weigand et al. 2014).

Taken together, our findings suggest that the observed effects are network-specific rather than global. We did not include these additional results in the revised manuscript because primary motor cortex was not in our initial contrast maps between probable AD and CN groups, which deviated from our overall study design. However, we will include these results in the manuscript if Reviewer #2 recommends otherwise.

**Author response image 1. sa2fig1:** Metabolic sensorimotor network and the network scores. A. Brain slice of metabolic covariance network associated with the primary motor cortex seed. B. The summary of individual-level metabolic network scores (mean and median) presented in half-violin plots. ‘*’ indicates significant group difference (p).

5. It seems odd that the CN group would have lower structural network scores than the MCI in the A-T- group. Please discuss this somewhat surprising finding.

We thank Reviewer #2 for raising this point. Although the CN group showed lower mean structural network scores than the MCI group in the A-T- group, this was not statistically significant when we performed a non-parametric one-way ANOVA test. Moreover, comparable medians of the two groups were demonstrated in the half-violin plots (see the revised *Figure 3B*). Therefore, we postulate that the relatively small sample size of the CN group and relatively younger mean age of MCI in the A-T- group may contribute to this finding. We did not mention this point in the main text as our manuscript focused mainly on statistically significant results.

6. There are many comparisons being done here (although it is unclear how many) – are p-values being corrected for multiple comparisons in Figures 2B and 3B? Please add these details.

We would like to clarify that the seeds used to define structural and metabolic networks were derived by comparing grey matter probability and glucose metabolic maps of probable AD and cognitively normal individuals. Our prior hypothesis was that brain networks defined based on these AD-related regions would show differences across different pathology groups and cognitive stages. As suggested by Reviewer #1 that non-parametric tests could increase the strength of the group comparison results, we have performed separate non-parametric one-way ANOVA analyses to test the differences between the cognitive stages and between the pathology groups for each network, based on a-priori hypotheses. Therefore, we did not perform multiple comparison correction for the number of networks analyzed, given that the analyses for the specific networks were hypothesis-driven rather than exploratory at the whole-brain level. Subsequently for the analyses that showed statistically significant group differences, we then performed post-hoc non-parametric tests between each pair of cognitive stages or pathology groups followed by Bonferroni-Holm correction for multiple comparisons (adjusted α = 0.05). We have added these details in the Statistical analyses sub-section under the Methods section.

7. There is likely a lot of correlation between the network measures (as evidenced by the fact that the network scores are very consistent across regions within the same group). It is known that LASSO will randomly suppress one of two correlated variables. Please use a different penalty (perhaps ridge) that does not have this drawback so that the interpretation of the coefficients can be more trustworthy.

We thank Reviewer #2 for raising this important point. As suggested, in addition to LASSO, we have performed a new analysis that used Ridge as the penalty in the SVC modelling. Using the ridge penalty method on the main dataset, we found very similar stage- and pathology-dependent associations of hippocampal structural and angular gyrus-based metabolic network scores with memory scores, consistent with our original analyses using LASSO. We have added these results into the main text (Results, pp. 15, paragraph 2), Figure 4—figure supplement 1D and Figure 5—figure supplement 1C.

8. Relatedly, Sup Figure 9 showing the variable selection frequency for permuted datasets is a bit confusing – the frequency ranges are 0 to 5 (unclear what unit this is representing), do not appear to be uniform/random and vary greatly between the main and validation dataset. I am not sure how these results support the main conclusions about the SVC model. Replication of the main findings using something other than Lasso/Elastic Net would be helpful to support the main findings.

We apologize for the confusion. To assess the specificity of the SVC results, we first randomly permuted the memory scores 100 times across the participants. For each of the 100 permuted datasets, we repeated the SVC modelling 100 times and selected the key predictors of memory using the brain measures that were consistently observed for more than 90 repetitions. This result supported the high specificity of the original SVC models as (1) the selection frequency of these key predictors in the 100 permuted datasets was low (1~6 times out of 100) and random, and (2) the hippocampal structural and angular gyrus-based metabolic network scores selected based on the original dataset were not favored over other variables in the models derived from the permuted datasets. This observation was true for both the main and validation dataset 1.

We have reported the total frequency of being selected as key predictors from the 100 permuted datasets of each brain measure in Figure 5—figure supplement 4 (re-numbered in the revised manuscript) and added the details in the legend of Figure 5—figure supplement 4. As suggested by Reviewer #2, we found similar findings using Ridge as the alternative penalty to LASSO in the SVC models. Moreover, we also performed the specificity tests using Ridge penalty on the permuted data sets. Similar to LASSO, the ‘null’ permutations using Ridge also yielded unbiased selection of predictors which supported the specificity of our findings (Figure 5—figure supplement 4).

9. The replication study is a bit confusing. It appears that 468 individuals were added to the original set of 812 to obtain a set of 1280 individuals, 859 of which were used in the final step of the analysis. Unless there is a misunderstanding, doesn't this mean that the "replication dataset" is actually mostly the original dataset? it would be stronger to replicate the study using only the 468 independent individuals – otherwise it is not a replication study.

We thank Reviewer #2 for this pertinent comment and agree that it will be stronger to replicate the findings using only data from the 468 independent individuals. However, in the original version, we noticed that a large number of these 468 subjects did not have tau and amyloid-β information around the same visit of MR scan (unlike those in the main dataset), thus the samples for the sparse varying coefficient (SVC) modelling in each pathology group were limited. That is why we decided to combine the main dataset with these new 468 subjects into a set of 1280 subjects to perform validation in the previous version.

In the revised manuscript, as suggested, we repeated the same analyses on these 468 subjects, i.e., validation dataset 2, including seed definition, network derivation using PLS, and brain-memory association trajectory using SVC modelling. Specifically, (1) we defined the seeds based on the group comparisons between the 152 CN and 190 probable AD individuals from validation dataset 2. (2) To increase the sample size as much as possible for each pathology group, we searched the ADNI database again and included 234 subjects who have the tau information within one year from the scan visit and amyloid-β information within three years from the scan visit. We then performed PLS analysis on these 234 subjects to derive the individual structural and metabolic network scores. (3) Lastly, we performed SVC modelling on the A+T-/A+T+ group (32 CNs, 39 MCIs, and 91 probable ADs).

Our key findings from the main dataset in A+T-/A+T+ group are replicated in the independent validation dataset 2 (Figure 4—figure supplement 1A). Specifically, hippocampal structural network score and angular gyrus-based metabolic network score remained as key predictors for memory performance. Furthermore, the trajectories of the associations between the brain network scores and memory performance were largely similar in both the main dataset and validation dataset 2. Unfortunately, we were unable to perform stage-dependent SVC modelling on the A-T- (27 CNs, 9 MCIs and 4 ADs) and A-T+ groups (22 CNs, 5 MCIs and 5 ADs) due to the small sample size. Nevertheless, our findings in all three pathology groups remained based on validation dataset 1, which combines the main and the validation dataset 2.

Taken together, we have demonstrated the replicability of our key findings. We have also included this as a potential future work in the Discussion (pp. 21, paragraph 1). We have added these findings in the main Results (pp. 14, paragraph 1), Supplementary Methods, Figure 1—figure supplement 1 and Figure 4—figure supplement 1.

References

Canal-Garcia, A., E. Gómez-Ruiz, M. Mijalkov, Y.-W. Chang, G. Volpe, J. B. Pereira and A. s. D. N. Initiative (2022). "Multiplex Connectome Changes across the Alzheimer’s Disease Spectrum Using Gray Matter and Amyloid Data." Cerebral Cortex.

DuPre, E. and R. N. Spreng (2017). "Structural covariance networks across the life span, from 6 to 94 years of age." Network Neuroscience 1(3): 302-323.

Lowe, V. J., S. D. Weigand, M. L. Senjem, P. Vemuri, L. Jordan, K. Kantarci, B. Boeve, C. R. Jack, D. Knopman and R. C. Petersen (2014). "Association of hypometabolism and amyloid levels in aging, normal subjects." Neurology 82(22): 1959-1967.

Spreng, R. N., E. DuPre, J. L. Ji, G. Yang, C. Diehl, J. D. Murray, G. D. Pearlson and A. Anticevic (2019). "Structural covariance reveals alterations in control and salience network integrity in chronic schizophrenia." Cerebral Cortex 29(12): 5269-5284.

Veldsman, M., H.-J. Cheng, F. Ji, E. Werden, M. S. Khlif, K. K. Ng, J. K. Lim, X. Qian, H. Yu and J. H. Zhou (2020). "Degeneration of structural brain networks is associated with cognitive decline after ischaemic stroke." Brain communications 2(2): fcaa155.